# Convergent transcriptomic signature in iPSC-dopaminergic neurons of hereditary Parkinson's disease

Irina V Kopylova[1,2,*], Artem B Ivanov[1,3,*], Lev R Eidelman[3], Ekaterina N Zaitseva[3], Ekaterina D Kulikova[3], Dmitriy A Grehnyov[4], Alexandra N Bogomazova[1,2], Vladimir A Vigont[4], Elena V Kaznacheyeva[4], Maria A Lagarkova[1,2], Olga S Lebedeva[1,2,†], Evgenii I Olekhnovich[1,2,†]

**The clinical and genetic diversity of Parkinson's disease (PD) makes it challenging to identify common mechanisms across different forms. To search for such shared pathways, we performed transcriptomic analysis of iPSC-derived dopaminergic neurons from patients with *LRRK2* or *Parkin* mutations. We discovered a convergent gene expression signature in both genetic backgrounds, indicating a shift away from developmental and proliferative programs (e.g., Wnt/$\beta$-catenin signaling, cell cycle) and toward pathways of mature neuronal function (e.g., synaptic transmission, potassium channels). This shift was particularly pronounced in *LRRK2* neurons, which showed enhanced markers of synaptic maturation. Concurrently, PD neurons exhibited down-regulation of gene programs supporting axon growth and structural development and upregulated TRAIL (TNF-related apoptosis-inducing ligand) apoptotic pathway. Our findings suggest that in these hereditary forms of PD, distinct mutations may propel neurons toward a similar state of premature specialization coupled with impaired structural development and increased vulnerability, revealing a potential common path in disease pathogenesis.**

## Introduction

Parkinson's disease (PD) is a severe neurodegenerative disorder with a steadily increasing prevalence (GBD 2016 Parkinson's Disease Collaborators, 2018). The core motor symptoms of PD result from the neurodegeneration of dopaminergic neurons (DAns) in the substantia nigra pars compacta and are often accompanied by a range of non-motor features, including rapid eye movement sleep behavior disorder, cognitive decline, and depression (Bloem et al, 2021). To date, no disease-modifying therapies are available, a failure largely attributable to our incomplete understanding of PD pathogenesis (Wu et al, 2025).

PD is a multifactorial disorder, where genetic predisposition, cell-intrinsic vulnerability of DAns, environmental factors, and aging converge (Johnson et al, 2019). Pathogenic mutations in at least 24 genes have been linked to the disease, many of which implicate key cellular pathways such as mitophagy (e.g., *PINK1*, *Parkin*) and the autophagy-lysosomal system (e.g., *GBA1*, *LRRK2*, *VPS35*) (Funayama et al, 2023). Despite this convergence on specific cellular functions, which suggests shared origins, the precise mechanisms linking these diverse defects to the core PD phenotype have yet to be elucidated. Mutations in *LRRK2* and *Parkin* represent two of the most common genetic causes of PD, yet they are associated with divergent clinical profiles, hinting at potentially distinct molecular mechanisms. *Parkin*-related PD is characterized by a significantly earlier onset and a more variable presentation of motor signs, whereas *LRRK2*-associated PD typically manifests later in life with a more consistent and penetrant motor syndrome (Kasten et al, 2018; Krüger et al, 2025). This clinical disparity positions these forms as critical, contrasting models for investigating the spectrum of molecular dysfunction in PD.

To identify shared mechanisms in PD, we leveraged patient-specific induced pluripotent stem cell (iPSC)-derived DAns. This model provides a renewable source of biologically relevant human cells for modeling disease-associated mechanisms at early developmental stages (Nolbrant et al, 2017; Lebedeva et al, 2023). Transcriptomic profiling of *LRRK2*- and *Parkin*-mutant DAns revealed a convergent signature. This signature reflects a coherent transcriptional shift: down-regulation of cell cycle maintenance

---

[1]Lopukhin Federal Research and Clinical Center of Physical-Chemical Medicine of Federal Medical Biological Agency, Moscow, Russia [2]Center for Genetic Reprogramming and Gene Therapy, Lopukhin Federal Research and Clinical Center of Physical-Chemical Medicine of Federal Medical Biological Agency, Moscow, Russia [3]ITMO University, St. Petersburg, Russia [4]Institute of Cytology, Russian Academy of Sciences, St. Petersburg, Russia

Correspondence: jeniaole01@gmail.com
*Irina V Kopylova and Artem B Ivanov contributed equally to this work
†Olga S Lebedeva and Evgenii I Olekhnovich contributed equally to this work

---

 

and developmental Wnt/β-catenin signaling pathway, coupled with a reciprocal upregulation of transcriptomic programs governing terminal neuronal specialization. Importantly, this shift toward specialization occurs with the suppression of structural development pathways. Collectively, these findings define a convergent signature in hereditary PD, where distinct mutations potentially drive neurons toward a transcriptomic state of precocious specialization paired with architectural deficiency.

# Results

## Characterization of iPSC-derived dopaminergic neurons from PD patients and controls

We generated iPSC-derived dopaminergic neurons (DAns) independently in two laboratories, using cells from four Parkinson's disease (PD) patients (carrying mutations in *LRRK2* or *Parkin*) and two healthy donors (Table 1), following a published protocol (Lebedeva et al, 2023). It should be noted that the patient with the G2019S mutation in *LRRK2* was also a carrier of the N370S variation in *GBA1*, which introduced a potential genetic confounding effect. Since we consider this phenotype to be primarily determined by the G2019S substitution in *LRRK2*, from now on we refer to the patient as "*LRRK2*." This is further elucidated in the limitations section. Cells were harvested at day 52–54 of differentiation for transcriptomic analysis (Fig 1). The cohort's composition included technical variables: one control line was from a female donor (others were male), and two *Parkin*-mutant lines were derived using lentiviral reprogramming, while all other lines used non-integrating Sendai virus.

This study analyzed 34 transcriptomic profiles from iPSC-derived models, comprising samples from four patients with PD (6 *LRRK2* and 16 *Parkin* samples) and two healthy donors (12 samples). Samples were sourced from two independent research centers (10 from Moscow [FRCC PCM], 24 from St. Petersburg [INC RAS]). After adapter trimming and quality filtering, we retained a total of ~723 million high-quality sequencing reads, corresponding to an average of 21 ± 9 million reads per sample (Table S1). We generated a raw count matrix through gene quantification (input reads per sample are shown in Fig S1). A two-way ANOVA on these raw counts was performed to evaluate technical batch effects, which identified the laboratory of origin as a significant source of variance (adj. $P < 0.0001$). Raw counts were normalized using the DESeq2 median-of-ratios method for all downstream comparative analyses (Table S2). To explore the global transcriptomic structure, principal component analysis (PCA) was performed on variance-stabilized transformed (VST) data (Fig S2A and B). Subsequent Permutational Multivariate Analysis of Variance (PERMANOVA) quantified variance contributions and demonstrated that the biological condition (disease status and mutation) was the strongest driver of global expression profiles ($R^2 = 0.26$, adj. $P < 0.0001$). Significant, though smaller, variance components were associated with technical covariates: laboratory of origin ($R^2 = 0.23$, $P < 0.0001$), donor gender ($R^2 = 0.12$, $P < 0.0001$), and reprogramming method ($R^2 = 0.07$, $P = 0.0002$). In contrast, technical replicates showed no significant effect ($P > 0.05$), confirming high experimental reproducibility.

Transcriptomic analysis validated the identity and revealed key phenotypic features of the iPSC-derived neuronal models (Fig S3A and B). First, our differentiation protocol successfully guided the cells to a complete exit from pluripotency, confirmed by the absence of core markers *NANOG* and *POU5F1*. The cells concurrently adopted a neural progenitor identity, evidenced by robust expression of *SOX2* (6,198 ± 1,671 reads). Expression of glial markers *GFAP* and *OLIG2* was also detected at lower levels (458 ± 1,104 and 296 ± 307 reads, respectively). Notably, expression of all three lineage markers — *SOX2* (Wilcoxon rank-sum test, $P = 4.64 \times 10^{-5}$), *GFAP* ($P = 0.0003$), and *OLIG2* ($P = 1.53 \times 10^{-8}$) — was significantly elevated in the St. Petersburg differentiation experiment compared to Moscow. We next identified significant, mutation-associated differences. Notably, expression of the dendritic maturation marker *MAP2* was elevated in *LRRK2* carriers compared to controls ($P = 2.12 \times 10^{-5}$), whereas *TUBB3* levels were unchanged. Conversely, proliferation markers *MKI67* and *TOP2A* were significantly down-regulated in the *LRRK2* group ($P < 0.05$). Comparative analysis between the two differentiation experiments revealed systematic differences. Samples generated in St. Petersburg exhibited higher expression of both proliferation markers (*MKI67*, *TOP2A*) and neuronal maturity markers (*MAP2*, *TUBB3*) than those from Moscow ($P < 0.0001$ and $P = 7.626 \times 10^{-9}$/$P = 0.03$, respectively). Finally, we confirmed that the models robustly expressed core ventral midbrain dopaminergic identity markers (*TH*, *FOXA2*, *LMX1A*, *OTX2*). A mixed-effects model indicated that their overall expression did not differ significantly by disease status, with the exception of *LMX1A*, which was differentially expressed in the *Parkin* group relative to controls ($P = 0.03$). Notably, the baseline expression of these markers varied between the two differentiation protocols: *OTX2* expression was significantly higher in the St. Petersburg cohort ($P = 0.01$), while *FOXA2* expression was elevated in the Moscow cohort ($P = 0.003$).

To evaluate the effectiveness of our previously published protocol for differentiating iPSCs into DAns (Lebedeva et al, 2023), the PDP1.5L cell line (as a representative) was characterized using immunocytochemical and flow cytometry analysis (Fig 2). Immunocytochemical staining revealed cells with normal neuronal morphology, expressing tyrosine hydroxylase (TH+) and βIII-tubulin (TUBB3+), and exhibiting long neurites (Fig 2A), confirming successful differentiation into the target phenotype. Quality control by flow cytometry on day 52 of differentiation demonstrated a reproducibly high proportion of TH+ cells across all experimental groups (Fig 2B). Representative overlaid histograms show the specific signal from anti-TH antibody staining compared to the isotype control. Further cell cycle analysis of the DAn populations on days 52–54 revealed a uniform profile characteristic of predominantly post-mitotic cells, with no significant proliferative differences between conditions (Fig 2C). Collectively, these data confirm the effective and reproducible generation of DAns using the presented protocol.

## Transcriptional signatures reveal abnormalities of neuronal differentiation in PD

Building on these findings, we hypothesized that *LRRK2* mutations disrupt the maturation dynamics of DAns. This premise was

**Table 1.  Overview of cell lines and sample numbers.**

| Cell line | Gender | Age | Reprogramming method | Mutation | Reference | Places and sample numbers |
|---|---|---|---|---|---|---|
| RG4S (Po4S) | M | 60 | Sendai virus | No | Holmqvist et al (2016) | Four from INC RAS, two from Lopukhin FRCC PCM |
| FF9S | F | 48 | Sendai virus | No | Bogomiakova et al (2021) | Four from INC RAS, two from Lopukhin FRCC PCM |
| PDL1.6S (BL6S) | M | 60 | Sendai virus | Mutation in the *PARK8 (LRRK2)* gene encoding the LRRK2 kinase (substitution 6055G>A [G2019S] in exon 41, the second mutation 1226A<G [N370S] in the *GBA1* gene) | Holmqvist et al (2016) | Four from INC RAS, two from Lopukhin FRCC PCM |
| PDP1.5L (Tr5L) | M | 66 | Lentivirus | Mutations in two alleles of the *PARK2 (Parkin)* gene encoding the E3-ubiquitin ligase parkin deletion 202–203 AG in the second exon and a splicing mutation in 1 intron (IVS1+1G/A) | Lebedeva et al (2023) | Four from INC RAS |
| PDP2.1L (P12-1) | M | 54 | Lentivirus | Heterozygous duplication of the second exon of the *PARK2 (Parkin)* gene | Gerasimova et al (2023) | Four from INC RAS, two from Lopukhin FRCC PCM |
| PDP4.4S (Park14 cl4) | M | 40 | Sendai virus | Heterozygous deletion of the second exon of the *PARK2 (Parkin)* gene | Shuvalova et al (2020) | Four from INC RAS, two from Lopukhin FRCC PCM |

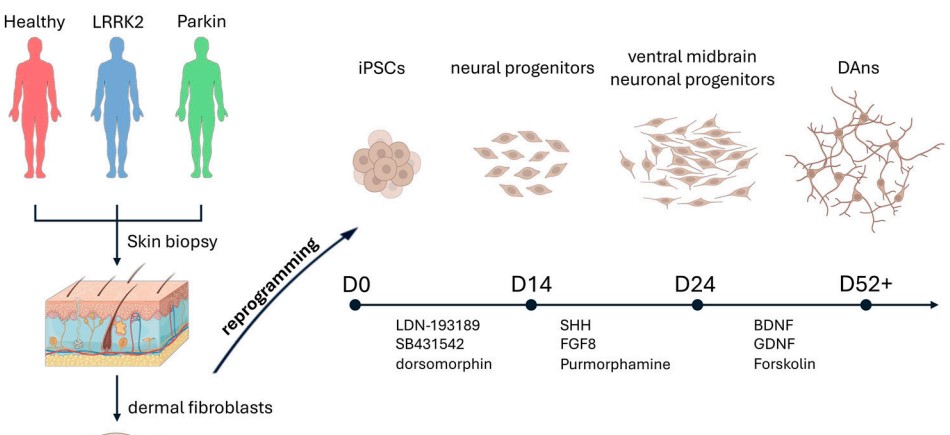

**Figure 1.  Scheme of differentiation of iPSCs into DAns and characteristics of cells used for transcriptome analysis.**
The experimental design included the production of iPSCs, their differentiation, and subsequent characterization and transcriptome analysis. Images adapted from Servier Medical Art (https://smart.servier.com), licensed under CC BY 4.0 (https://creativecommons.org/licenses/by/4.0/).

supported by previous findings indicating that *LRRK2*-mutant DAns exit the cell cycle prematurely, which may impact their long-term viability (Walter et al, 2021). To investigate whether a similar effect was present in our models, we employed a comparative transcriptomic analysis leveraging publicly available bulk RNA-seq data from Linker et al (2022). This external dataset profiled pan-neuronal differentiation across a temporal series, capturing transcriptomes from neural progenitors (baseline) and at 2, 4, 6, and 8 wk of differentiation (metadata in Table S3, raw counts in Table S4). PCA of the reference dataset revealed a distinct separation along PC1, segregating neuronal progenitors from all neuronal samples at later time points (2, 4, 6, and 8 wk), which clustered together (Fig S4A and B). Subsequent differential expression analysis with thresholds of FDR < 0.05 and |log$_2$FC| > 1 identified consistent marker genes that distinguished progenitors from differentiating neurons across all time points. From this,

we defined two core gene signatures: a "neurons" cluster comprising 468 genes with progressively increased expression during differentiation, and a "progenitors" cluster of 258 genes with higher expression in the progenitor state (Table S5).

To annotate the distinct molecular signatures associated with the cellular identity in our dataset, we performed an over-representation analysis (ORA). This revealed a clear dichotomy between enriched pathways characteristic of mature neuronal function and those governing cell proliferation (Table S6). The "neurons" gene set was overwhelmingly enriched for core synaptic and electrophysiological functions. The most significant terms were associated with the synaptic membrane (GO:0097060; enrichment ratio, ER = 6.00, FDR = 4.05 × 10$^{-22}$) and the neuronal system pathway (R-HSA-112316; ER = 5.76, FDR = 1.36 × 10$^{-21}$). We further observed strong enrichment for the regulation of trans-synaptic signaling (GO:0099177; ER = 4.86, FDR = 1.72 × 10$^{-17}$), neuron-

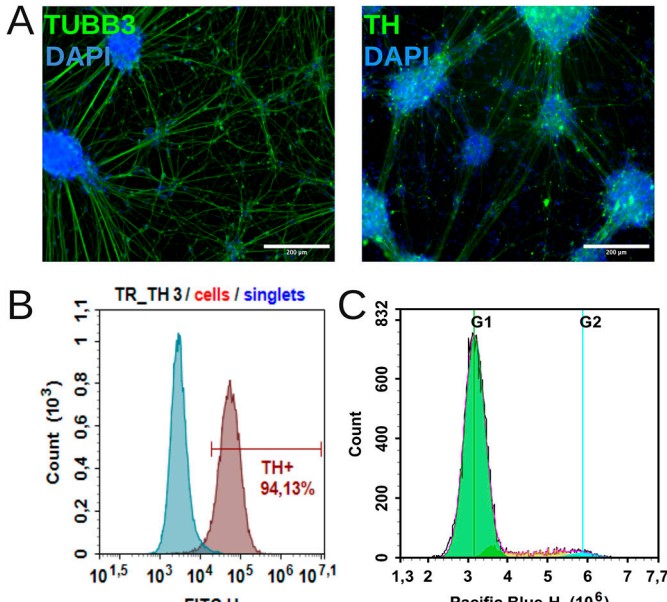

**Figure 2. Validation of dopaminergic neuron differentiation protocol.**
**(A)** Representative immunocytochemical staining of DAns on day 52 of differentiation; TH and TUBB3 – green, DAPI – blue; scale bar 200 *μm*.
**(B)** Representative flow cytometry analysis of DAns on day 52 of differentiation. The data are presented as overlaid histogram plots of anti-TH antibody-stained cells (brown) and isotype control antibody-stained cells (blue).
**(C)** Representative graph showing cell-cycle analysis of DAns at day 52–54 of differentiation.

to-neuron synapse (GO:0098984; ER = 5.44, FDR = 1.01 × 10⁻¹⁶), and postsynaptic specialization (GO:0099572; ER = 5.48, FDR = 4.78 × 10⁻¹⁶). Underlying this synaptic specialization was a coordinated enrichment for ion channel activity, including gated channel activity (GO:0022836; ER = 4.99, FDR = 6.17 × 10⁻¹²) and regulation of membrane potential (GO:0042391; ER = 4.36, FDR = 1.19 × 10⁻¹²). In stark contrast, the "progenitors" gene set was dominated by terms related to active cell division. The most profoundly enriched pathways were the general cell cycle (R-HSA-1640170; ER = 9.49, FDR = 2.02 × 10⁻⁶³) and Cell Cycle, Mitotic (R-HSA-69278; ER = 10.09, FDR = 7.76 × 10⁻⁵⁶). This signature pointed to active genomic replication and segregation, with high enrichment for DNA replication (GO:0006260; ER = 12.00, FDR = 2.55 × 10⁻³⁵), chromosome segregation (GO:0007059; ER = 11.11, FDR = 1.13 × 10⁻⁴⁹), and organelle fission (GO:0048285; ER = 9.68, FDR = 7.18 × 10⁻⁴⁵). Key regulatory mechanisms were also highlighted, including Cell Cycle Checkpoints (R-HSA-69620; ER = 12.42, FDR = 3.18 × 10⁻³⁹) and the regulation of cell cycle phase transition (GO:1901987; ER = 10.13, FDR = 4.88 × 10⁻⁴¹). This ORA confirms the expected biological dichotomy in our samples: one gene signature reflects neuronal communication machinery, while the other is indicative of a proliferative, progenitor-like state.

To delineate transcriptomic profiles associated with PD and specific genetic mutations, we performed differential gene expression analysis. We identified differentially expressed genes (DEGs) for four primary group comparisons using thresholds of FDR < 0.05 and |log₂FC| > 1. The resulting DEGs were used to create

ranked lists for subsequent Gene Set Enrichment Analysis (GSEA) with our custom "neurons" and "progenitors" signatures as gene sets. GSEA revealed that the "neurons" signature, associated with neuronal specialization, was significantly enriched in both the *LRRK2* and *Parkin* groups compared to healthy controls, while the "progenitors" signature was significantly depleted (|NES| > 2, *P*.adj < 0.001). Furthermore, a direct comparison between the *LRRK2* and *Parkin* groups showed a similar pattern: the "neurons" signature was enriched (NES = 1.19, *P*.adj = 0.03), and the "progenitors" signature was depleted (NES = −1.77, *P*.adj < 0.001) in *LRRK2* neurons relative to *Parkin* neurons. These results potentially suggest that DAns from both *LRRK2* and *Parkin* carriers exhibit transcriptional profiles indicative of an accelerated differentiation and maturation trajectory compared to controls, with *LRRK2* neurons demonstrating a more pronounced shift than *Parkin* neurons. These findings are summarized in Fig 3.

### Identification and validation of condition-specific transcriptomic signatures

To identify common and mutation-specific transcriptional signatures, we first analyzed the PD-linked DEG lists generated in the prior analysis. An initial examination revealed substantial overlap among these gene sets (Fig S5). To isolate unique, condition-specific signatures, overlapping genes were sequentially removed. This refinement process yielded final, non-overlapping DEG lists for each core condition compared to healthy controls: 414 genes for the general PD cohort (144 up-regulated, 270 down-regulated), 1,644 genes for *LRRK2* carriers (503 up-regulated, 1,141 down-regulated), and 220 genes for the *Parkin* group (124 up-regulated, 96 down-regulated). The composition of these unique signatures is summarized in a final intersection plot (Fig 4A), with complete gene lists provided in Table S7. Furthermore, given the unbalanced experimental design, we separately identified DEGs associated with key technical covariates — laboratory of origin (2,619 up-regulated, 1,926 down-regulated), donor gender (1,884 up-regulated, 1,762 down-regulated), and reprogramming method (1,034 up-regulated, 948 down-regulated) — using the same statistical thresholds (Tables S8, S9, and S10, respectively).

To provide an initial biological interpretation of these signatures, we performed exploratory ORA, which revealed contrasting molecular landscapes (Table S11). In the general PD cohort, primary disturbances were associated with core neuronal functions, as evidenced by significant enrichment (FDR < 0.25) in pathways regulating neurotransmitter transport (GO:0006836; enrichment ratio (ER) = 7.11, FDR = 0.12) and potassium ion transport (GO:0006813; ER = 6.28, FDR = 0.14), including potassium channels (R-HSA-1296071; ER = 10.10, FDR = 0.14). In contrast, the healthy control signature was characterized by the predominance of developmental and morphogenic signaling pathways, most notably the Hippo signaling pathway (hsa04390; ER = 7.49, FDR = 0.001) and cell-cell signaling by Wnt (GO:0198738; ER = 4.21, FDR = 0.001). The *LRRK2*-associated profile suggested a dual pathology. Up-regulated genes were strongly associated with synaptic function, showing profound enrichment in the regulation of trans-synaptic signaling (GO:0099177; ER = 3.95, FDR = 2 × 10⁻⁸) and synapse organization (GO:0050808; ER = 3.30, FDR = 2 × 10⁻⁵). Conversely, down-

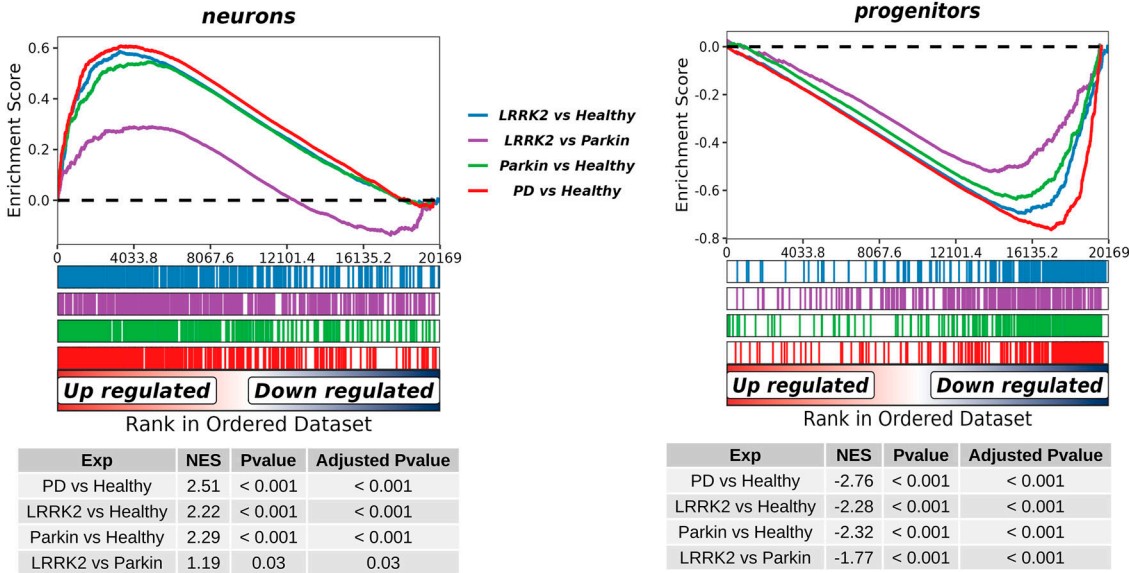

**Figure 3.  Gene Set Enrichment Analysis (GSEA) of differentiation signatures in iPSC-derived DAns.**
Enrichment plots for the custom "neurons" (top) and "progenitors" (bottom) gene signatures across key group comparisons. Colored bars and enrichment scores (ES) indicate the direction and significance of enrichment for each signature when comparing *LRRK2* or *Parkin* groups to healthy controls, and *LRRK2* directly to *Parkin*.

regulated genes exhibited a coordinated suppression of ECM components, supported by highly significant terms for ECM organization (R-HSA-1474244; ER = 3.25, FDR = 1 × 10$^{-5}$) and collagen chain trimerization (R-HSA-8948216; ER = 8.57, FDR = 1 × 10$^{-5}$). The *Parkin*-associated signature implicated a different mechanism centered on signaling dysregulation. The upregulated profile (FDR < 0.25) included a single significant term for the regulation of insulin-like growth factor (IGF) transport and uptake (R-HSA-381426; ER = 11.49, FDR = 0.19). Meanwhile, the down-regulated genes were significantly involved in G-protein-coupled receptors (GPCR) ligand binding (R-HSA-500792; ER = 6.30, FDR = 0.01) and the ERK1 and ERK2 cascade (GO:0070371; ER = 10.20, FDR = 0.0001) - pathways linked to the positive regulation of cytosolic calcium ion concentration (GO:0007204; ER = 11.83, FDR = 0.01).

## Systems-level protein interaction networks reveal coherent dysregulated modules

To move beyond pathway-centric lists and elucidate the functional architecture of the identified consistent DEGs (FDR < 0.05), we constructed a stepwise protein-protein interaction (PPI) network. This systems-level approach independently validated and refined the initial ORA findings by mapping DEGs onto highly coherent interaction clusters. The analysis revealed distinct, condition-specific PPI modules that link each genetic background to cohesive biological programs. DEGs associated with the PD state were enriched within a voltage-gated potassium channel complex cluster (59 genes). In contrast, DEGs linked to the healthy control profile formed clusters annotated for canonical Wnt signaling (46 genes) and chromosome segregation (41 genes). Notably, these same healthy-associated pathways — canonical Wnt signaling (NES = −1.59, *P*.adj = 0.03) and chromosome segregation (NES = −1.48, *P*.adj = 0.03) — were also significantly reduced in the

*Parkin* group when compared directly with the *LRRK2* group. This suggests a greater dysregulation of these developmental and homeostatic pathways in *LRRK2*-associated pathology relative to *Parkin* mutations. Further stratification by genetic background revealed an additional layer of functional specificity. In *LRRK2*-associated models, upregulated DEGs coalesced into a large cluster governing trans-synaptic signaling (n = 234), while down-regulated DEGs were partitioned into modules for interferon signaling and ECM-receptor interactions (n = 60). Parallel analysis of *Parkin*-associated models showed that down-regulated DEGs were significantly enriched within a GPCR signaling cluster (n = 72). These dense, functionally coherent PPI clusters are visualized in Fig 4B–D. The biological relevance of these network modules was rigorously validated using GSEA, which confirmed their significant and systematic association with the respective experimental groups. This confirms that the expanded PPI clusters represent functionally coherent and biologically dysregulated pathways in our iPSC-derived neuronal models. The complete gene lists for each STRING cluster, their intersections with the original DEGs, and detailed genes included to PPI clusters and ORA/GSEA results are provided in Tables S12, S13, and S14 and Figs S6A and B and 4E.

## Systematic pathway enrichment validates divergent molecular programs in LRRK2 and parkin PD

To deepen and confirm our previous results, we performed a classical GSEA using multiple databases, including MsigDB (HALLMARK), gene ontology (GO), KEGG, and Reactome (FDR < 0.05; Table S15). A consistent signal across different comparison groups was observed exclusively in the HALLMARK, GO, and Reactome results. Our analysis, consistent with the differential gene expression data, revealed both common and unique molecular pathways for each mutation. The intersections of these differential

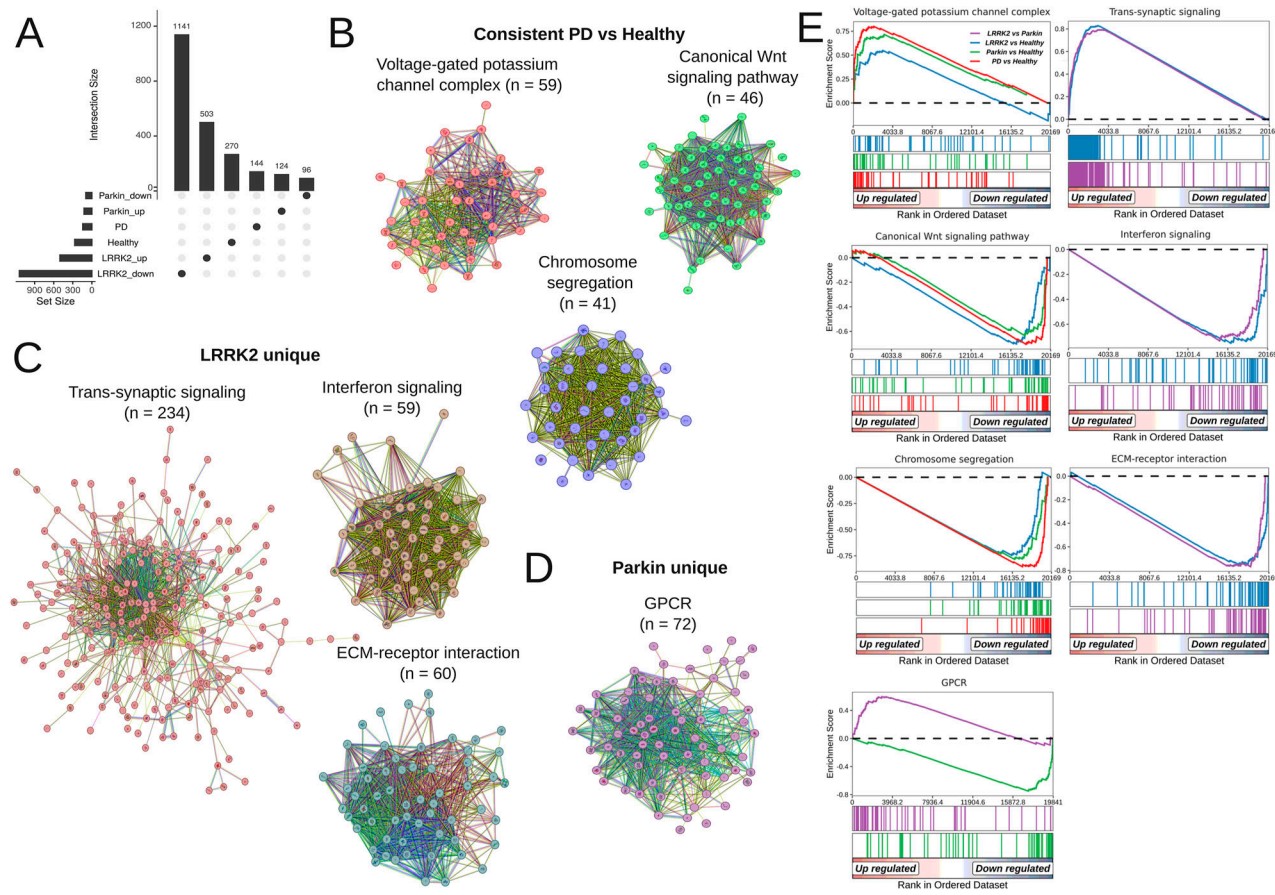

**Figure 4. Distinct molecular signatures and functional networks of PD genetic subtypes.**
**(A)** Upset plot confirming minimal overlap between refined DEG sets after removal of common genes. **(B, C, D)** Condition-specific protein-protein interaction networks showing key functional clusters for (B) PD-associated, (C) *LRRK2*-associated, and (D) *Parkin*-associated DEGs. Networks were constructed from the consistent DEGs for each condition using the STRING database v12.0, with edges representing both physical and functional associations (combined score > 0.4). Dense functional clusters were identified using the MCL algorithm and annotated via pathway enrichment analysis. Cluster size is indicated in brackets. **(E)** GSEA validation of protein-protein interaction clusters across experimental comparisons. Enrichment Scores with statistical significance are shown for each cluster. Color bars represent different group contrasts (PD versus healthy, *LRRK2* versus healthy, *Parkin* versus healthy, *LRRK2* versus *Parkin*).

pathways are visualized in Fig S7A–D, while enrichment plots for pathways classified as unique or common to *LRRK2* and *Parkin* are provided in Fig S8. A consistent pattern in PD group was characterized by an up-regulation of transmission across chemical synapses (R-HSA-112315), coupled with a healthy group-linked HALLMARK E2F targets, homology directed repair through homologous recombination (R-HSA-5685942), and cell cycle checkpoints (R-HSA-69620). The *LRRK2* mutations were uniquely associated with the upregulation of postsynaptic and axonal components, such as postsynaptic density membrane (GO: 0098839), distal axon (GO:0150034), and regulation of synapse organization (GO:0050807), as well as class A/1 (rhodopsin-like receptors) (R-HSA-373076). Conversely, it exhibited a pronounced down-regulation of ECM-related processes, including ECM organization (GO:0030198) and extracellular structure organization (GO: 0043062). In contrast, the *Parkin* mutation displayed a unique down-regulation of pathways involved in cellular response to starvation (R-HSA-9711097), epigenetic regulation of gene expression (R-HSA-212165), regulation of endogenous retroelements

(R-HSA-9842860), and NGF-stimulated transcription (R-HSA-9031628). An antiphase relationship was observed for ribosomal and translational processes. Pathways, such as influenza infection (R-HSA-168255), influenza viral RNA transcription and replication (R-HSA-168273), cytosolic ribosome (GO:0022626), eukaryotic translation termination (R-HSA-72764), response of EIF2AK4 (GCN2) to amino acid deficiency (R-HSA-9633012), and nonsense-mediated decay (R-HSA-927802), were up-regulated in *Parkin* compared to both *LRRK2* and healthy controls, but were conversely down-regulated in *LRRK2* under the same comparative conditions.

## Confounding signatures of place, gender, and reprogramming methodology

### *Place*

Differentiation performed in Moscow was specifically associated with the strong up-regulation of pathways related to protein maturation, secretory trafficking, and metabolic adaptation, as revealed by GSEA; we considered pathways with a Normalized

Enrichment Score, NES > 2. The signature was overwhelmingly dominated by processes related to endoplasmic reticulum function and stress. This included the most prominent finding, protein processing in endoplasmic reticulum (KEGG: NES = 2.26, $P$.adj = $1.24 \times 10^{-12}$), and a battery of related GO biological process terms, such as response to endoplasmic reticulum stress (NES = 2.23, $P$.adj = $4.84 \times 10^{-15}$) and endoplasmic reticulum unfolded protein response (NES = 2.18, $P$.adj = $5.12 \times 10^{-9}$). This biosynthetic and stress-adaptive signature was mechanistically supported by the upregulation of vesicular transport pathways, including COPII-coated endoplasmic reticulum to Golgi transport vesicle (GO cellular component: NES = 2.09, $P$.adj = $2.62 \times 10^{-7}$) and Golgi vesicle transport (GO BP: NES = 2.07, $P$.adj = $9.17 \times 10^{-12}$). Concurrently, we observed a significant metabolic reprogramming, characterized by the enrichment of glycolytic process (NES = 2.02, $P$.adj = $5.28 \times 10^{-6}$) and the hallmark hypoxia pathway (NES = 2.28, $P$.adj = $2.60 \times 10^{-13}$), alongside the activation of nucleotide and sugar metabolism pathways (e.g., amino sugar and nucleotide sugar metabolism, KEGG: NES = 2.05, $P$.adj = $3.21 \times 10^{-5}$).

In contrast, differentiation performed in St. Petersburg was characterized by a pronounced signature of cell proliferation, genome maintenance, and transcriptional regulation. The most significant findings were a strong enrichment for E2F Targets (hallmark: NES = −2.73, $P$.adj = $8.60 \times 10^{-41}$) and the G2/M checkpoint (NES = −2.69, $P$.adj = $2.36 \times 10^{-37}$), indicating a transcriptional program driving cell cycle progression. This signature aligns with the profound and coordinated down-regulation of core cell cycle and chromosome maintenance pathways observed in the GO analysis, including chromosome organization (NES = −2.65, $P$.adj = $7.84 \times 10^{-51}$), chromosome segregation (NES = −2.64, $P$.adj = $3.22 \times 10^{-44}$), and cell cycle phase transition (NES = −2.40, $P$.adj = $4.91 \times 10^{-32}$). This proliferative state was further supported by the upregulation of the cell cycle (KEGG: NES = −2.59, $P$.adj = $3.02 \times 10^{-25}$) and mitotic spindle (hallmark: NES = −2.40, $P$.adj = $5.61 \times 10^{-20}$) pathways. Furthermore, extensive activity was observed in DNA repair mechanisms, including homologous recombination (KEGG: NES = −2.11, $P$.adj = $4.22 \times 10^{-6}$), highlighting a state of active genome surveillance, which corresponds to the suppression of double-strand break repair (GO: NES = −2.52, $P$.adj = $5.60 \times 10^{-29}$) and DNA recombination (NES = −2.42, $P$.adj = $1.39 \times 10^{-23}$) in the contrasting signature. Finally, the upregulation of RNA splicing (KEGG spliceosome: NES = −2.09, $P$.adj = $5.87 \times 10^{-7}$) and Chromatin organization (reactome: NES = −1.75, $P$.adj = $7.53 \times 10^{-5}$) points to a tightly regulated epigenetic and post-transcriptional landscape.

GSEA comparing differentiation samples from Moscow and St. Petersburg revealed a striking reciprocal pattern. The "neurons" signature was significantly enriched in the Moscow group (NES = 1.84, FDR = $1.250215 \times 10^{-11}$), while the "progenitors" signature was strongly depleted in the same group (NES = −2.95, FDR = $5.02 \times 10^{-13}$). Based on the pronounced enrichment of cell cycle, DNA repair, and chromatin organization pathways, combined with the GSEA results for the maturation signatures, we propose that the St. Petersburg differentiation experiment maintains a larger, more active pool of neural progenitors. In contrast, the Moscow experiment appears to drive cells on average further along the differentiation trajectory, yielding a population with a stronger mature neuronal signature and a correspondingly smaller progenitor compartment.

Collectively, these data capture two divergent in vitro phenotypes: one enriched cells in the proliferation stage (St. Petersburg), and another advanced toward neuronal maturation (Moscow).

### Gender

To identify gender-associated molecular signatures, we performed GSEA using the Hallmark MSigDB gene set collection. This revealed a pronounced sexual dimorphism in the baseline state of the control cell lines. The male RG4S line exhibited a signature of active cell proliferation. Specifically, we observed coordinated upregulation of the G2/M checkpoint pathway (NES = 1.63, $P$.adj = 0.027) and E2F target genes (NES = 1.57, $P$.adj = 0.027) compared to the female FF9S line. This proliferative state was further corroborated by a significant up-regulation of our custom "progenitors" signature in the male line (NES = 1.91, FDR = $8.88 \times 10^{-8}$), coupled with a down-regulation of the "neurons" signature (NES = −2.00, FDR = $5.02 \times 10^{-13}$). Conversely, the female FF9S line displayed a distinct profile characterized by the up-regulation of immune-inflammatory and stress-response pathways. The most prominent enrichments included the inflammatory response (NES = 1.67, $P$.adj = 0.024), TNFA signaling via NFKB (NES = 1.56, $P$.adj = 0.033), and the p53 pathway (NES = 1.59, $P$.adj = 0.027). Notably, the detection of up-regulated cell-cycle pathways in the male control line is significant. While their down-regulation has been previously linked to PD in male-only cohorts, our finding effectively excludes the possibility that such a result was an artifact of using a female control line for comparison.

### Reprogramming

To identify transcriptional signatures inherent to the reprogramming process, we performed GSEA on pathway gene sets from the GO, Hallmark MSigDB, reactome databases (Table S16). A pronounced Sendai virus-specific signature was identified, characterized by the significant up-regulation of key biological processes. The most prominent finding was a strong activation of epithelial mesenchymal transition (NES = 1.93, $P$.adj = 0.0019). This was accompanied by the up-regulation of interferon alpha response (NES = 1.85, $P$.adj = 0.031) and complement pathway signaling (NES = 1.72, $P$.adj = 0.035). In addition, processes related to coagulation (NES = 1.78, $P$.adj = 0.035) and myogenesis (NES = 1.66, $P$.adj = 0.035) were enriched, painting a coherent picture of Sendai virus-induced remodeling. Despite this strong Sendai signature, ECM pathways emerged as a primary point of divergence between methods. Lentiviral reprogramming induced a significant down-regulation of ECM pathways relative to Sendai virus reprogramming, evidenced by strong enrichment scores for the ECM organization pathway (reactome, NES = 1.94, $P$.adj = 0.00034) and the ECM cellular component (GO, NES = 2.05, $P$.adj = $6.43 \times 10^{-7}$). Notably, this lentivirus-associated ECM-down-regulated state converged with the transcriptional profile of *LRRK2*-mutant lines. Despite being generated with the Sendai virus method — which upregulates ECM — the *LRRK2*-mutant lines exhibited a similarly suppressed ECM signature. This convergence suggests that both the lentiviral vector and the *LRRK2* mutation ultimately impact the same regulatory networks governing ECM expression. In parallel, our analysis revealed a significant down-regulation of core epigenetic regulatory pathways in lentivirus reprogrammed cells, notably in

chromatin organization (NES = −1.69, P.adj = 0.0050) and epigenetic regulation of gene expression (NES = −1.48, P.adj = 0.030). Intriguingly, this epigenetic suppression signature converges with the unique transcriptional profile previously attributed to *Parkin* mutations, suggesting a shared disruption of chromatin-related expression. As two of the three *Parkin*-mutant lines in our study were generated through lentivirus reprogramming, this convergence makes it challenging to delineate the specific epigenetic effects of the mutation from those inherent to the reprogramming process itself. In addition, it is worth noting that the G alpha (i) signaling events pathway is down-regulated in the lentivirus reprogramming group (NES = −1.80, P.adj = 0.03). This observation is critical for interpretation, as the broader GPCR pathway was also found to be down-regulated in *Parkin* carriers. The concurrent down-regulation in both the lentivirus group and the *Parkin* signature suggests that this specific transcriptomic effect cannot be attributed solely to the *Parkin* mutation and is likely confounded, at least in part, by the reprogramming method. Furthermore, we identified a pronounced and method-specific signature of cytoskeletal remodeling. A divergence was observed in pathways governing cilium organization and microtubule-based motility, which were up-regulated in Sendai virus-reprogrammed lines but down-regulated in lentiviral-reprogrammed counterparts. This opposing regulation was underscored by the significant enrichment of key processes including axoneme assembly (GO:0035082, NES = 2.16, P.adj = 0.0013), microtubule-based movement (GO:0007018, NES = 1.84, P.adj = 0.0013), and motile cilium function (GO:0031514, NES = 1.91, P.adj = 0.0079).

**PD-related and differentiation-specific co-expression modules**

***PD-related co-expression modules***
To identify functionally coordinated gene clusters and investigate potential regulatory mechanisms in DAns carrying *LRRK2* and *Parkin* mutations, we constructed a co-expression network. Prior to analysis, batch effects associated with the place were successfully mitigated. The effectiveness of this correction was confirmed by PERMANOVA, which showed that variance linked to the place was negligible (R$^2$ = 0.01, P = 0.98), while the variance explained by the condition or mutation remained significant (R$^2$ = 0.12, P = 0.0003). Network analysis revealed specific gene modules with significant associations to the experimental groups. Using Pearson correlation, the skyblue3 (83 genes), black (92 genes), and gray (479 genes) modules exhibited significant positive correlations with the healthy group and negative correlations with the PD group. Furthermore, these same modules were significantly negatively correlated with the *Parkin* mutation group. A separate analysis using Spearman correlation identified the skyblue3 module (102 genes) as also being positively correlated with the healthy group and negatively correlated with both the PD and *Parkin* groups. The complete gene lists for each co-expression module are provided in Table S17. Heatmaps with information about links co-expression modules with experimental groups presented in Fig S9A.

Finally, pairwise correlations between the eigengene expression of all identified modules were found to be statistically significant (Spearman correlation, FDR < 0.05), indicating a coordinated higher-order structure within the gene co-expression network. The expression of module eigengenes demonstrated significant differences between the PD and healthy control groups, as determined by a generalized linear model (FDR < 0.05; skyblue-pearson: 0.019, black: 0.006 gray: 0.03 skyblue-spearman: 0.006). Given the significant co-expression and correlation between these modules themselves, this coordinated down-regulation suggests the presence of a concerted transcriptional program that becomes dysregulated with PD mutations. Selected eigenegenes across experimental groups presented in Fig S9B.

To elucidate the functional landscape of gene co-expression networks linked to PD, we performed ORA on the identified modules. The analysis revealed distinct yet complementary biological themes pointing to processes critical for neural development and cellular structure. The statistical signals were observed in the gray module and the cross-module ORA analysis of all genes. The gray module was highly and significantly enriched for pathways related to cell adhesion and cytoskeletal organization, including cell junction assembly (GO:0034329; ER = 3.17, FDR = 0.0000039), adherens junction (GO:0005912; ER = 4.62, FDR = 0.0000039), and actin binding (GO:0003779; ER = 3.09, FDR = 0.0000068). This theme was corroborated by the analysis of all genes, which also identified cell junction assembly (GO:0034329; ER = 3.08, FDR = 0.00000002) and actin binding (GO:0003779; ER = 3.05, FDR = 0.000000041) as the top hits, underscoring the fundamental role of these processes in the overall transcriptomic profile. The remaining modules also showed associations with pathways involved in nervous system development; however, it is characterized by few overlapping genes and/or insufficient statistical significance. Finally, the integrative ORA analysis of all genes confirmed the convergence of these major themes, identifying a core set of significantly enriched pathways that include nervous system development (R-HSA-9675108; ER = 2.62, FDR = 0.00000015), axonogenesis (GO:0007409; ER = 2.90, FDR = 0.00000010), and VEGFA-VEGFR2 signaling (WP3888; ER = 2.71, FDR = 0.0000035). In summary, functional enrichment analysis delineates a network where modules are associated with biological processes, primarily focused on neural development, axon guidance, and the underlying cellular structures of adhesion and the ECM. The overall ORA results presented in Table S18.

Pathway activity analysis and co-expression modules revealed distinct patterns. PCA of the four module eigengenes indicated that the first principal component 1 explained over 75% of the total variance. The expression profile captured by PC1 showed a significant positive correlation with TGF-*β* pathway activity (Spearman's *ρ* = 0.57, FDR = 0.01). Notably, however, a direct comparison of TGF-*β* pathway activity levels across the healthy, *LRRK2*, and *Parkin* groups did not yield statistically significant differences. In contrast, two other pathways exhibited significant alterations in the mutation groups compared with healthy controls, as shown in Fig 5. The TNF-related apoptosis-inducing ligand (TRAIL) apoptotic pathway was significantly up-regulated (*LRRK2*: FDR = 0.005; *Parkin*: FDR = 0.00007), while the Wnt signaling pathway was significantly down-regulated (*LRRK2*: FDR = 0.04; *Parkin*: FDR = 0.02). Importantly, in the linear model used for this analysis, the covariates of gender and reprogramming method were not significant.

***Differentiation site specific co-expression modules***
To characterize patterns associated with the site of differentiation, we performed a comparative analysis of location-specific co-expression

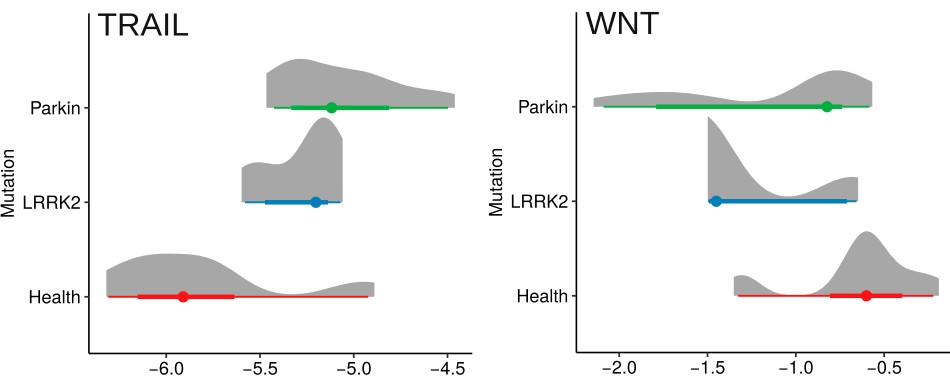

**Figure 5. Pathway activity analysis reveals a convergent dysregulation pattern in PD.**
Bar plots show significant up-regulation of the TRAIL apoptotic pathway (*LRRK2* FDR = 0.005; *Parkin* FDR = 0.00007) and down-regulation of WNT signaling (*LRRK2* FDR = 0.04; *Parkin* FDR = 0.02) in mutant neurons compared to healthy controls. Significance was determined using a generalized linear model with FDR correction.

modules linked to Moscow and St. Petersburg differentiation experiments. This analysis successfully delineated several key modules within each group using both Pearson and Spearman correlation methods. For the Moscow group, four distinct modules were identified: three via Pearson correlation (darkturquoise, grey 60, and violet) and one via Spearman correlation (lightsteelblue1). For the St. Petersburg group, six modules were defined, comprising three Pearson-based modules (cyan, lightcyan1, and orangered4) and three Spearman-based modules (gray, midnightblue, and orange). Detailed gene lists for all identified co-expression modules may be consulted in Table S19.

Exploratory ORA of gene co-expression modules, generated using Pearson and Spearman correlation methods and filtered for signals, revealed distinct molecular landscapes between Moscow and St. Petersburg cohorts. The signatures from the Moscow group, notably from the darkturquoise (Pearson) and lightsteelblue1 (Spearman) modules, were dominated by terms related to fundamental neuronal architecture and synaptic communication. This included profound enrichment for the distal axon (GO:0150034; ER = 8.12, FDR = $1.2 \times 10^{-4}$), vesicle-mediated transport in synapse (GO:0099003; ER = 13.85, FDR = $7.6 \times 10^{-6}$), and the presynaptic active zone (GO:0048786; ER = 28.06, FDR = $7.6 \times 10^{-6}$). These findings were supported by significant involvement of core pathways such as the neuronal system (R-HSA-112316; ER = 6.05, FDR = $4.3 \times 10^{-6}$) and axon guidance (R-HSA-422475; ER = 4.98, FDR = $6.7 \times 10^{-6}$). Conversely, the St. Petersburg group's signature, prominently featured in the orangered4 (Pearson) module, highlighted a strong association with tissue development and structural organization. The most significant enrichments pointed to processes of ossification (GO:0001503; ER = 7.65, FDR = $2.8 \times 10^{-7}$), ECM organization (R-HSA-1474244; ER = 6.85, FDR = $1.3 \times 10^{-3}$), and skeletal system morphogenesis (GO:0048705; ER = 7.62, FDR = $2.3 \times 10^{-3}$). A separate, large gray (Spearman) module in the same cohort also showed significant enrichment for neuronal development processes, including axonogenesis (GO:0007409; ER = 3.17, FDR = $1.4 \times 10^{-5}$) and synapse organization (GO:0050808; ER = 2.91, FDR = $6.5 \times 10^{-5}$). The detailed results of ORA presented in Table S20.

Our earlier functional enrichment analysis delineated a location-specific divergence in the underlying transcriptional programs. The signature of the Moscow group was enriched for pathways characteristic of mature neurons, whereas the St. Petersburg group profile was elevated in progenitor-associated

signatures, indicating an earlier state of neuronal maturation. Consequently, the co-expression modules identified in the Moscow cohort are linked to the specialized physiology of mature neurons, while those from the St. Petersburg group reflects earlier stages of neuronal differentiation and maturation. This spectrum of transcriptional states suggests a coordinated molecular program underlying neuronal maturation. Notably, the modules significantly associated with both PD groups — skyblue3 (Pearson), black (Pearson), gray (Pearson), and skyblue3 (Spearman) — occupy an intermediate position along this axis (Fig 6). The correlation pattern between pathway activities and module eigengenes mirrored the inter-module correlations themselves. This coherence confirms that these transcriptional programs are coordinated and are likely governed by distinct upstream signaling pathways (Fig S10).

It is worth noting that GSEA revealed an increase in PD-related modules within neurons whose signature shifted toward greater specialization. This increase was statistically significant in the following comparisons: between the Moscow and St. Petersburg samples (module skyblue3: Pearson NES = 1.61, FDR = 0.008; Spearman NES = 1.46, FDR = 0.02) and, in the data from Linker et al (2022), between third-party data and week 4 of differentiation (module black, Pearson: NES = 1.59, FDR = 0.013), week 6 (module black, Pearson: NES = 1.49, FDR = 0.04; module skyblue3, Spearman: NES = 1.4, FDR = 0.04), and week 8 (module skyblue3, Pearson: NES = 1.44, FDR = 0.03; module black, Pearson: NES = 1.47, FDR = 0.03; module skyblue3, Spearman: NES = 1.46, FDR = 0.03). This pattern suggests that as PD-associated neurons undergo greater specialization, they may lose a specific, coordinated functional program that is typically present. Given that our prior functional enrichment implicated these PD-associated modules in fundamental neuronal development pathways, we hypothesize that the affected cells exhibit defects not only in maturation but also in developmental growth, likely due to dysregulation of programs essential for neuronal structure.

# Discussion

PD is a progressive neurodegenerative disorder classically characterized by the loss of DAns in the substantia nigra. While aging remains the primary risk factor, the etiology of PD is complex,

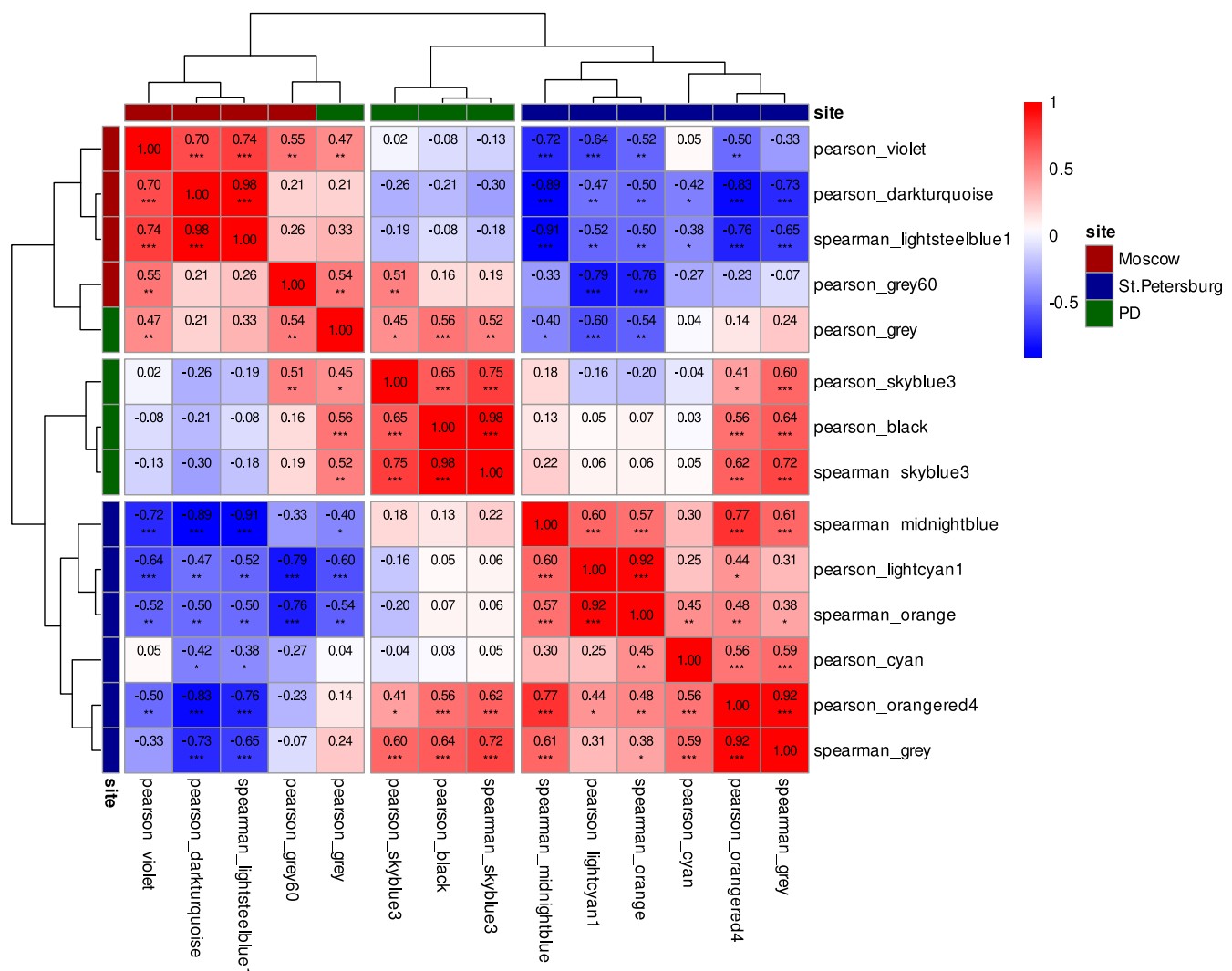

**Figure 6. Location-specific co-expression signatures and their relationship to PD-associated modules.**
Pairwise correlations of eigengenes from co-expression modules associated with PD and differentiation location. The heatmap depicts Spearman correlation coefficients, with positive correlations shown in red and negative correlations in blue. Coefficient values are displayed within each cell. Statistical significance is denoted by asterisks: ***$P$ < 0.001, **$P$ < 0.01, *$P$ < 0.05.

involving both genetic susceptibility and environmental influences (Pang et al, 2019). Increasingly, evidence points to early neuro-developmental alterations as a potential source of lifelong vulnerability, suggesting that pathogenic processes may begin decades before clinical manifestation. Subtle deviations in brain development can establish aberrant developmental patterns that manifest later in life(Walter et al, 2021). Our study builds upon this premise by employing a transcriptomic profiling of human iPSC-derived DAns — a model well-suited for investigating early human neuro-development (Mariani et al, 2012). Notably, the rejuvenation inherent to reprogramming, which often limits the ability to model aging (Studer et al, 2015), is advantageous here for probing developmental mechanisms. This direction is supported by prior findings that the *LRRK2*-G2019S mutation accelerates cell-cycle exit and dopami-nergic differentiation via *NR2F1* down-regulation, concurrently in-creasing cell death (Walter et al, 2019; Nickels et al, 2020; Walter et al,

2021). To systematically map shared and mutation-specific patho-logical pathways underlying such neurodevelopmental deficits, we analyzed transcriptomic data from *LRRK2*- and *Parkin*-associated models. Our analysis revealed a core pathological signature com-mon to both genetic forms, characterized by the coordinated suppression of cell cycle-related pathways (Wnt/$\beta$-catenin signal-ing, chromosome segregation, cell-cycle checkpoints) alongside the activation of pathways governing neuronal specialization (voltage gated potassium channel, transmission across chemical synapses). We propose that this transcriptional pattern provides direct mo-lecular evidence for accelerated differentiation and premature cell-cycle exit in DAns derived from both *LRRK2* and *Parkin* groups, a mechanism that may ultimately contribute to reduced mature neuronal viability (Walter et al, 2021).

Integrally linked to the observed signature of accelerated dif-ferentiation is the coordinated down-regulation of Wnt/$\beta$-catenin

signaling pathways, a hallmark of the healthy control signature that is lost in our PD models. This finding presents two nonexclusive interpretations. On one hand, the suppression of Wnt/$\beta$-catenin signaling could represent a necessary, programmed step in neuronal maturation, facilitating the transition from a proliferative or plastic state to a fully differentiated, post-mitotic neuron. On the other hand, and more compelling in the context of PD, this down-regulation may constitute a core pathological mechanism. We hypothesize that mutations in *LRRK2* and *Parkin* induce a pathogenic state that actively dysregulates the Wnt/$\beta$-catenin pathway. This aberrant suppression would then derail normal differentiation, compromising the formation of a resilient, mature neuronal phenotype and contributing to premature cellular vulnerability. The central role of Wnt/$\beta$-catenin signaling in both development and maintenance of DAns is well-established. Foundational studies have shown it is critical for the specification and neurogenesis of midbrain DAns from the floor plate (Joksimovic & Awatramani, 2014). Crucially, its function extends into adulthood, where it acts as a master regulator of neural stem cell dynamics, synaptic integrity, and neuronal identity in neurogenic niches (Marchetti et al, 2020). In PD and aging, this vital signaling is suppressed through multiple mechanisms, including increased activity of the inhibitor GSK-3$\beta$, up-regulation of endogenous antagonists (e.g., DKK1), and a decline in supportive Wnt ligands from astrocytes (L'Episcopo et al, 2014; Marchetti et al, 2020). This dysfunction is exacerbated by chronic neuro-inflammation, which creates a hostile microenvironment that further inhibits neurogenic and repair capacities (L'Episcopo et al, 2014). Consequently, the suppression of these crucial pathways in our PD models indicates a profound failure in key maintenance and resilience programs. It suggests a pathological regression of cellular identity and cripples essential pro-repair mechanisms, severely limiting endogenous neuroplasticity. This mechanistic understanding underscores the Wnt/$\beta$-catenin pathway as a promising therapeutic target, with strategies to reactivate it showing potential to stimulate neurorepair in PD models (Marchetti et al, 2020).

In addition to the down-regulation of developmental and pro-survival Wnt/$\beta$-catenin pathway, our analysis uncovered a convergent proapoptotic signature across both genetic backgrounds. The TRAIL pathway was up-regulated in PD neurons compared with healthy controls. TRAIL is a well-characterized inducer of extrinsic apoptosis, and its up-regulation in post-mitotic DAns suggests that these cells acquire an intrinsic predisposition to cell death already at early stages of differentiation (Tisato et al, 2016). The concurrent activation of TRAIL-mediated death signaling together with the suppression of Wnt/$\beta$-catenin and the impairment of structural support programs likely creates a permissive state in which neurons become exquisitely sensitive to later age-related stressors. These findings add a further layer to the concept of developmental vulnerability in PD, showing that both the dismantling of pro-survival pathways and the active priming of apoptotic machinery occur early in the disease course.

Furthermore, our analysis revealed that co-expression modules which associated cell junctions, neuronal development, and axonogenesis were down-regulated in PD. Furthermore, the expression patterns of these modules were highly correlated with

each other and, collectively, showed a significant positive correlation with TGF-$\beta$ signaling activity. This suggests that the down-regulation of these structural and developmental programs in PD may be functionally linked to alterations in the TGF-$\beta$ pathway. This connection is biologically plausible, as TGF-$\beta$ signaling is a well-documented regulator of pro-survival functions, synaptic integrity, axonal guidance, and ECM dynamics in neurons (Yi et al, 2010; Hinz, 2015; Deng et al, 2024). We propose this regulatory interaction may be mediated epigenetically. Studies in other systems show that HDAC1, HDAC2, and HDAC3 can control TGF-$\beta$-dependent gene expression and cellular transitions (Barter et al, 2010; Thambyrajah et al, 2018). Critically, the pathogenic relevance of HDAC dysregulation in PD is underscored by work in iPSC-derived dopamine neurons, where nuclear mislocalization of HDAC4 acts as an upstream regulator of a progressive disease axis culminating in ER stress (Lang et al, 2019). In our model, despite finding no significant difference in predicted TGF-$\beta$ ligand levels between groups, the coordinated down-regulation of its target program persists. This points to an upstream disruption, likely via HDACs, that potentially uncouples the normal transcriptional output from TGF-$\beta$ signaling.

Beyond the common core signature, neurons carrying the *LRRK2* G2019S mutation exhibited a distinct transcriptomic profile, characterized by the pronounced up-regulation of genes mediating specialized neuronal functions. This perspective is reinforced by findings that *LRRK2*-G2019S accelerates dopaminergic differentiation through down-regulation of the key developmental transcription factor *NR2F1*, suggesting its hyperactivity directly perturbs neurodevelopmental programs (Walter et al, 2021). Specifically, the up-regulated pathways included those involved in trans-synaptic signaling, distal axon function, postsynaptic density membrane organization, and the regulation of synapse organization. This observed transcriptomic shift is directionally and mechanistically supported by recent studies elucidating *LRRK2*'s direct role in synaptic biology, consistent with its known functions in regulating synaptogenesis. Pathogenic LRRK2 hyperphosphorylates the GTPase RAB3A, impairing its interaction with the motor adapter MADD and disrupting the anterograde axonal transport of synaptic vesicle precursors to pre-synaptic sites (Dou et al, 2024). Concurrently, LRRK2 functions as a key scaffold in BDNF-triggered signaling, orchestrating actin-dependent postsynaptic remodeling and spinogenesis; its loss abolishes BDNF-induced synaptic plasticity in human neurons (Tombesi et al, 2024). Taken together, these independent lines of evidence support the mechanistic hypothesis that gain-of-function mutations in LRRK2 drive an accelerated neuronal maturation process, which in turn leads to structural deficits and the consequent aberration of both presynaptic and postsynaptic function.

While *LRRK2* mutations converge on synaptic dysfunction, our analysis suggested that the *Parkin*-associated profile may implicate a distinct mechanism centered on epigenetic dysregulation, characterized by a unique down-regulation of pathways involved in epigenetic gene regulation, control of endogenous retroelements, and NGF-stimulated transcription. However, a critical methodological caveat tempers this interpretation. The same pathways were significantly down-regulated in association with the lentiviral reprogramming method used for two of the three *Parkin* lines. Therefore, while intriguing, this specific epigenetic signature requires validation in models free of such technical

artifacts. Nonetheless, the overarching theme of epigenomic ab-errations remains a compelling and convergent vulnerability in PD pathogenesis. This concept is strongly supported by external ev-idence beyond our potentially confounded signature (Madabhushi et al, 2014; Fernández-Santiago et al, 2015; Ainslie et al, 2021).

Thus, this accelerated shift toward a specialized state repre-sents an early developmental defect — a maladaptive tran-scriptomic transition that drives functional maturation while failing to establish the necessary supportive cellular architecture. This mismatch likely creates a lasting neuronal vulnerability, laying the mechanistic groundwork for the subsequent synaptic, mito-chondrial, and other cellular dysfunctions observed across the PD spectrum (Labbé et al, 2016; Chen et al, 2022). Our integrated transcriptomic analysis of iPSC-derived DAns harboring either the *LRRK2* G2019S or *Parkin* mutations reveals a fundamental con-vergence on a neurodevelopmental deficit, despite the starkly different initial molecular lesions — constitutive kinase activation versus ubiquitin ligase loss-of-function. The convergent signature, characterized by up-regulated neuronal specialization programs alongside suppressed Wnt/β-catenin signaling and neuro-developmental pathways, provides direct molecular evidence that distinct genetic causes of PD converge on a shared pathogenic mechanism. This subtle, initial deviation in differentiation dy-namics may represent the deterministic "butterfly effect" of monogenic mutations. While not immediately catastrophic, this early vulnerability likely lowers the threshold for subsequent age-related insults, such as oxidative stress, proteostatic decline, or neuroinflammation, to trigger overt neurodegeneration and the classic motor manifestations decades later.

## Limitations

This study is subject to several methodological limitations. First, the *LRRK2* group consisted of neurons derived from a single donor, which limits our ability to draw broader conclusions about *LRRK2*-associated pathology. Future studies with expanded patient cohorts are needed to validate these findings. However, our observation of accelerated differentiation in *LRRK2* neurons aligns with prior research reporting premature cell cycle exit in *LRRK2*-mutant DAns, supporting the biological relevance of our results (Walter et al, 2021).

Second, the co-occurrence of *LRRK2* G2019S and *GBA1* N370S mutations in the studied case introduces potential genetic con-founding. PD is increasingly recognized as a syndrome of diverse etiology, and the pathophysiological processes triggered by spe-cific genetic mutations remain poorly defined. When interpreting the *LRRK2*-specific signature, we assumed the phenotype was predominantly driven by the *LRRK2* G2019S mutation, given that the N370S variant in *GBA1* is associated with milder pathogenicity, lower penetrance, and the majority of its carriers do not develop the condition (Anheim et al, 2012; Zhang et al, 2024). This as-sumption is supported by previous data indicating a dominant association of *LRRK2* over *GBA1* (Ortega et al, 2021). Crucially, however, the primary aim of this study was to identify a convergent pathological signature across distinct genetic forms of PD. Therefore, the transcriptional disruptions found to be shared between this complex *LRRK2*/*GBA1*-mutant line and the *Parkin*-mutant lines are likely to represent core, disease-relevant

pathways rather than artifacts of the coincident *GBA1* mutation. Nonetheless, the precise biological underpinnings of the inter-action between these mutations remain obscure, and we cannot exclude a contributory or modifying role of the *GBA1* variant on the overall transcriptional profile.

Third, the experimental approach employed iPSC-derived DAns in monoculture, a system that inherently recapitulates only cell-autonomous pathological mechanisms. This model excludes the critical contribution of non-cell-autonomous processes mediated by glial populations, particularly astrocytes and microglia, which have established roles in neuroinflammatory cascades subse-quent to neuronal dysfunction (di Domenico et al, 2019; Bailey & Cookson, 2024; Trainor et al, 2024).

Fourth, our cohort exhibited inherent technical and demo-graphic heterogeneities that constitute confounding variables. It is known that the reprogramming method can influence the tran-scriptome of terminal iPSCs (Churko et al, 2017) and possibly differentiated cells. Notably, the choice of reprogramming method was identified as a driver of transcriptional variation, affecting pathways related to the ECM and epigenetic regulation. It is im-portant to note that this technical signature was identified through internal comparisons within the small *Parkin* cohort itself, which consisted of only three lines: one reprogrammed using the Sendai method and two using lentiviral vectors. Furthermore, the specific *Parkin* mutations carried by these lines differed, which may dif-ferentially impact neuronal biology. Thus, distinguishing between the residual effects of the reprogramming method and the au-thentic, mutation-specific consequences remains exceptionally challenging. This convergence makes it difficult to disentangle mutation-specific effects or individual Sendai-reprogramming PDP4.4S line from persistent technical signatures, and the ob-served patterns therefore require validation in future studies designed to explicitly control for these variables. Moreover, we encountered a similar issue when accounting for the gender variable in our analysis. While we identified transcriptomic sig-natures that may potentially be associated with gender, our comparisons involved only one cell line derived from a male donor (RG4S) and one from a female donor (FF9S). Although this ap-proach allowed us to control for effects specific to that particular female line and to mitigate potential gender-related confounding, we cannot definitively attribute these findings to biological gender. This is because, in such a limited comparison, it is impossible to disentangle the influence of gender from the individual genetic and epigenetic backgrounds of these specific donor-derived cell lines.

Finally, while we interpret the observed down-regulation of developmental pathways (e.g., Wnt/β-catenin signaling) and up-regulation of mature neuronal markers as evidence of premature or accelerated neuronal maturation, our experimental design cannot definitively rule out alternative cellular explanations. Specifically, the same transcriptional signature could, in principle, result from a shift in cellular composition within the cultures, such as an accelerated loss of neuronal progenitor populations relative to more mature neurons. However, we consider this alternative explanation of selective cell death less plausible, as the observed transcriptomic signature featured a coordinated up-regulation of pro-survival and neuronal function pathways, not the broad

activation of stress responses typically associated with such a mechanism. Furthermore, we cannot exclude the possibility that these distinct cellular mechanisms, namely accelerated maturation or shifts in population dynamics, might have contributed differentially to the transcriptomic profiles of the *LRRK2* and *Parkin* models, potentially accounting for some of the mutation-specific signatures observed.

# Materials and Methods

## Reagents and materials

DMEM (PanEco), DMEM/F12 (GIBCO), mTeSR1 (Stem Cell Technologies), GibriS-8 (PanEco), FBS (HyClone), KnockOut serum replacement (GIBCO), Glutamax (GIBCO), Penicillin-streptomycin (PanEco), N2 supplement (GIBCO and PanEco), B27 supplement (GIBCO and PanEco), Neurobasal (GIBCO), Neurobasal-A (GIBCO).

## Recombinant proteins and small molecule compounds

SHH (Miltenyi biotec), FGF8 (Miltenyi biotec), BDNF (Miltenyi biotec), GDNF (Miltenyi biotec), Purmorphamine (Stem Cell Technologies), Forskolin (Stem Cell Technologies), SB431542 (Stem Cell Technologies), LDN-193189 (Stem Cell Technologies), Dorsomorphin (Stem Cell Technologies), ROCK inhibitor (Y27632) (Stem Cell Technologies), Ascorbic acid (Sigma-Aldrich).

## Other reagents for cell cultures

Other reagents for cell cultures are Versene solution (PanEco), Matrigel (Corning), DMSO (PanEco), Trypsin 0.05% (GIBCO).

## Molecular biology reagents

Molecular biology reagents are PBS (PanEco), PBS without $Ca^{2+}$ and $Mg^{2+}$ (PanEco), PFA (Sigma-Aldrich), triton X-100 (Sigma-Aldrich), Tween20 (Sigma-Aldrich), goat serum (Hyclone), DAPI (Sigma-Aldrich), ethanol (Merck), TRIzol (Invitrogen), Duplex-specific nuclease (Evrogen).

## Consumables

Cell culture treated dishes with diameter 35 mm, multi-well plates (Corning), centrifuge tubes, serological pipettes, cryovials (Costar).

## Cell culture media compositions

Neural differentiation medium: DMEM/F12, 2% serum replacement, 1% N2 supplement, 1× glutamax, 50 U/ml penicillin-streptomycin, 10 $\mu$M SB431542, 2 $\mu$M dorsomorphin and 0.5 $\mu$M LDN-193189. Neuronal progenitor medium: DMEM/F12, 2% B27 supplement, 1× glutamax, 50 unit/ml penicillin-streptomycin, 100 ng/ml SHH, 100 ng/ml FGF8, 2 $\mu$M purmorphamine and 200 $\mu$M ascorbic acid. NB maturation medium: neurobasal, 2% B27 supplement, 1× glutamax, 50 unit/ml penicillin-streptomycin, 20 ng/ml BDNF, 20 ng/ml GDNF, 200 $\mu$M ascorbic acid and 4 $\mu$M forskolin. NBA maturation medium: neurobasal-A, 2% B27 supplement, 1× glutamax, 50 unit/ml penicillin-streptomycin, 20 ng/ml BDNF, 20 ng/ml GDNF, 200 $\mu$M ascorbic acid and 10 $\mu$M forskolin.

## The experiment protocols

### iPSCs cultivation

This study included iPSCs from three patients with genetically determined autosomal recessive early-onset form of PD associated with mutations in *Parkin*; patient with substitutions G2019S in *LRRK2* and N370S in *GBA1*. Two lines of iPSCs from healthy donors were used as control lines. A detailed description of the iPSCs used in the work is given in Table 1. iPSCs were maintained in mTeSR1 or GibriS-8 media according to the manufacturer's instructions. Matrigel was used for coating culture plates. Preparation of Matrigel-coated culture dishes and plates was performed according to the manufacturer's instructions. Cells were passaged using 0.05% trypsin. ROCK inhibitor (Y27632) was added to the culture medium for 1 d after seeding at a concentration of 5 $\mu$M.

### iPSC differentiation to DAns

iPSCs were differentiated to DAns as described in Lebedeva et al (2023). Briefly, iPSCs were detached with trypsin, plated on multi-well plates coated with Matrigel and cultivated in mTeSR1 or GibriS-8 medium until reaching a density of about 90–100%. Then the growth medium was replaced with a neural differentiation medium, in which cells were cultured for the next 14 d. From this stage and until the end of differentiation, the medium was changed every other day. The neural progenitors were detached with Versene solution, plated at a density of 250,000–400,000 cells/cm$^2$ on multi-well plates coated with Matrigel and cultured for 10 d in neuronal progenitor medium. The resulting ventral midbrain neuronal progenitors were also detached with Versene solution, plated at a density of 400,000 cells/cm$^2$ on multi-well plates coated with Matrigel and cultured for 7 d in NB maturation medium and then for three more weeks in NBA maturation medium until reaching full maturity. To ensure high-quality biological replicates, the differentiation into DAns was performed independently in two laboratories (the Lopukhin FRCC PCM and the INC RAS) using identical culture media compositions and following the same protocol. Details on the number of samples for each cell line and their laboratory of origin are provided in Table 1. For each iPSC line, two to four independent technical replicates were generated from a single differentiation batch at each experimental site. RNA was isolated from each replicate and sequenced as an independent library.

### Immunocytochemical analysis

Cells prepared for immunostaining were washed two times with PBS, fixed with 4% PFA in PBS (pH 6.8) for 20 min and washed with PBS three times. Cell membranes were permeabilized using PBS solution with 0.1% Triton X-100 for 10 min. Nonspecific sorption of antibodies was blocked by incubation for 30 min in PBS with 0.1% Tween 20 containing 5% FBS and 2% goat serum. Primary antibodies against TH (ab112; Abcam) and TUBB3 (ab7751; Abcam) were diluted in PBS with 0.1% Tween 20 containing 5% FBS and 2% goat

serum according to the recommendations of the manufacturer. Cells were incubated overnight at +4°C and then washed three times for 5 min in PBS with 0.1% Tween 20. Secondary anti-mouse or anti-rabbit IgG antibodies (Invitrogen) conjugated with fluorescent labels (Alexa 488, Alexa 555) were applied in dilutions recommended by the manufacturer, incubated for 30 min at RT in the dark and washed three times for 5 min with PBS containing 0.1% Tween 20. After that, cells were incubated in 0.1 μg/ml DAPI in PBS for 10 min to visualize the cell nucleus and washed twice with PBS. Microphotographs of cells were obtained using an inverted fluorescence microscope Olympus IX53F (Olympus) with cellSens Standard Version 1.11 software (Olympus). Image processing was performed using open-source ImageJ Version 1.54p.

### Flow cytometry and cell cycle analysis

Mature neurons were carefully detached with trypsin, re-suspended in DMEM/F12 medium containing 10% FBS and centrifuged at 300$g$ for 5 min at 4°C. After that, cells were washed twice with PBS, centrifuged under the same conditions each time. Then, cells were re-suspended using 1% PFA and incubated on ice for 15 min, then washed twice with PBS without $Ca^{2+}$ and $Mg^{2+}$ supplemented with 1% FBS. After the last wash, 100 μl of washing solution was left, and the cell pellet was re-suspended in this volume. Next, 80% ethanol was carefully added to the cell suspension. After that, cells were incubated on ice for 30 min.

For immunostaining, cells were washed twice with PBS without $Ca^{2+}$ and $Mg^{2+}$ supplemented with 1% FBS. Before the last wash each sample was divided into two tubes. Pellets were re-suspended in 100 μl of primary antibody solution in PBS without $Ca^{2+}$ and $Mg^{2+}$ supplemented with 1% FBS. Antibodies against TH (ab112; Abcam) were added to one part of the sample while isotype control antibodies (026102; Invitrogen) were added to another. Samples were incubated on ice for 1 h. After that, cells were washed twice using PBS without $Ca^{2+}$ and $Mg^{2+}$ supplemented with 1% FBS. Secondary antibodies conjugated with Alexa 488 (Invitrogen) were added to cells and incubated on ice for 30 min. Cells were washed twice with PBS without $Ca^{2+}$ and $Mg^{2+}$ supplemented with 1% FBS, and then were re-suspended in the same buffer. An equal volume of 400 ng/ml DAPI solution was added. Samples were analyzed using an Acea NovoCyte 3000 flow cytometer (Agilent). The obtained data were analyzed using the NovoExpress software (Agilent).

For cell-cycle analysis, cells were washed twice with PBS without $Ca^{2+}$ and $Mg^{2+}$ supplemented with 1% FBS, and then re-suspended in proportion to 200,000 cells/100 μl in 1 μg/ml DAPI solution in PBS. After that, cells were incubated for 30 min on ice and used for cell-cycle analysis. Cell-cycle analysis was performed by an Acea NovoCyte 3000 flow cytometer using the automated cell-cycle analysis option of NovoExpress Software. The Watson model was used for cell-cycle fitting. The obtained data were analyzed using the NovoExpress software.

### RNA extraction and transcriptome library construction

This study analyzed 34 transcriptomic samples, comprising 22 from 4 PD patients — stratified into 6 *LRRK2* and 16 *Parkin* genotypes

— and 12 from 2 healthy donors. All cellular differentiations were conducted in two technical replicates, and the cohort was sourced from two independent laboratories (10 from Moscow, 24 from St. Petersburg). Following differentiation, total RNA was isolated from mature neurons using TRIzol reagent according to the manufacturer's protocol. The RNA was then treated with a recombinant duplex-specific nuclease to remove genomic DNA, and its integrity was verified by horizontal gel electrophoresis. Sequencing libraries were constructed from the high-quality RNA using the TruSeq mRNA Stranded Kit (New England Biolabs), which facilitated poly(A) RNA enrichment and cDNA synthesis. The resulting cDNA libraries were assessed for quality on a fragment analyzer system (Agilent) and quantified via qPCR. All sequencing libraries were prepared and subjected to paired-end sequencing (2 × 150 cycles) on an Illumina NovaSeq 6000 platform by Evrogen. To match the experimental design, sequencing was performed in two independent runs, corresponding to the samples derived from the two independent laboratories. The raw data were processed using bcl2fastq v2.20 conversion software (Illumina) to generate FASTQ files in Phred 33 format, yielding a total of 854,110,128 reads.

### Bioinformatics analysis

### Preprocessing and quantification of raw sequencing data

Raw transcriptomic data underwent quality assessment using FastQC v0.12.1, with summary reports consolidated via MultiQC v1.27.1 (Ewels et al, 2016). Adapter sequences and low-quality bases were trimmed using Fastp v0.23.4 (Chen et al, 2018). The resulting high-quality reads were aligned to the GRCh38.p14 reference genome using STAR v2.7.11 (Dobin et al, 2013) in two-pass mode, guided by the GENCODE v47 annotation. Key alignment parameters included — *outSAMtype BAM SortedByCoordinate and — outFilterIntronMotifs RemoveNoncanonical*. Finally, gene-level read counts were generated with HTSeq-count v2.0.5 (Anders et al, 2015).

### Derivation of neuronal differentiation signatures from reference data

Reference transcriptomes for neuronal differentiation states were obtained from a publicly available RNA-seq dataset (GEO: GSE120271, Linker et al, 2022), encompassing neuronal progenitors (day 0) and neuronal cultures at 2, 4, 6, and 8 wk of differentiation. Raw count data processed using DESeq2 v1.44.0 in R. PCA was performed on variance-stabilized transformed (VST) data to bi-dimensionnaly visualize transcriptomic data structure. To identify markers of differentiation, we performed pairwise differential expression analysis between the progenitor time point (day 0) and each later time point of differentiation (weeks 2, 4, 6, 8) using a threshold of adjusted $P$ value (FDR) < 0.05 and absolute $\log_2$ fold change (LFC) > 1. Genes consistently up-regulated across all four neuronal time points relative to progenitors were defined as the "neurons" signature. Genes consistently down-regulated across the same comparisons were defined as the "progenitors" signature. The biological affiliation of the defined signatures was assessed through ORA using the WebGestalt (Elizarraras et al, 2024). Gene ontology (GO) (The Gene Ontology Consortium, 2021)

and Reactome pathways (Milacic et al, 2024) were queried with significance filtered for a FDR < 0.05. To interrogate differentiation states in our iPSC-derived DANs, we applied GSEA. For each relevant group comparison (e.g., *LRRK2* versus Healthy, *Parkin* versus Healthy, *LRRK2* vs. *Parkin*), a pre-ranked list was generated by sorting all expressed genes by the product of the sign of their LFC multiplied by $-\text{Log}_{10}$ (*P* value). GSEA was then performed using the clusterProfiler v4.12.6 in R, using our custom neurons and progenitor signatures as gene sets. Enrichment results with a normalized enrichment score (NES) and FDR < 0.05 were considered statistically significant.

### Initial processing and exploratory analysis of PD transcriptomes

The overall data structure was visualized using PCA on the VST data. The statistical significance of the observed separations, driven by key experimental factors, was subsequently quantified using Permutation Multivariate Analysis of Variance (PERMA-NOVA). Differential expression analysis was performed using DESeq2 v1.44.0 (Love et al, 2014) with a design formula of ~place (St. Petersburg or Moscow) + replicate (1 or 2) + condition (healthy or PD) or mutation (healthy, *LRRK2*, or *Parkin*) to account for technical variation.

### Targeted expression analysis of lineage and maturation markers

To assess the cellular identity and differentiation status of the iPSC-derived models, a targeted analysis of lineage-specific marker genes was performed. Gene expression levels for key markers were extracted from the normalized transcriptomic data. Pluripotency was evaluated via expression of the stem cell marker *SOX2*, *POU5F1*, *NANOG*. Astroglial and oligodendroglial contamination was assessed using markers *GFAP* and *OLIG2*, respectively. Neuronal maturity was evaluated by analyzing expression of the canonical markers *MAP2* and *TUBB3* (also known as *βIII*-Tubulin). Proliferative capacity was inferred from the expression of cell-cycle markers *MKI67* and *TOP2A*. Statistical analysis of marker expression between experimental groups was conducted using linear mixed-effects models to account for the nested data structure. For each marker, expression was modeled as a function of the mutation group (fixed effect) with a random intercept for the laboratory of origin (place) to control for batch effects (expression ~ mutation + (1 | place)). Models were fitted using the restricted maximum likelihood (REML) method via the lmer function from the lmerTest package v3.1-3 in R. Significance of fixed effects was determined using Satterthwaite's method for degrees of freedom approximation. Pairwise post hoc comparisons between mutation groups and the control group were performed, with *P*-values adjusted for multiple testing where applicable. Model diagnostics included visual inspection of residual plots to verify assumptions of normality and homoscedasticity. To directly compare the proliferation marker expression between the two laboratories independent of genotype, a nonparametric Wilcoxon rank-sum test was applied to the pooled data.

### Identification of core DEG signatures

We tested specific linear contrasts for four pairwise comparisons: PD versus healthy controls, *LRRK2*-associated versus healthy, *Parkin*-associated versus healthy, and *LRRK2*-associated versus *Parkin*-

associated. Genes with an absolute LFC > 1 and FDR < 0.05 were classified as differentially expressed, where a positive LFC indicated higher expression in the first named group of a contrast. To identify consistent, condition-specific transcriptional signatures, we defined sets of consistent DEGs through a multi-step intersection strategy designed to isolate genes with reproducible associations. The core PD signature was derived from the intersection of genes commonly up-regulated in the PD versus healthy, *LRRK2* versus healthy, and *Parkin* versus healthy comparisons. Conversely, the healthy control signature was defined by the intersection of genes commonly down-regulated in those same three contrasts. For mutation-specific profiles, *LRRK2*-up-regulated genes were identified as the intersection of the up-regulated fractions from the *LRRK2* versus healthy and *LRRK2* versus *Parkin* comparisons, while *LRRK2*-down-regulated genes came from the intersection of the down-regulated fractions from those same two comparisons. The *Parkin*-up-regulated set was defined as the intersection of genes up-regulated in *Parkin* versus healthy and genes down-regulated in *LRRK2* versus *Parkin*. Finally, the *Parkin*-down-regulated set was defined as the intersection of genes down-regulated in *Parkin* versus Healthy and genes up-regulated in *LRRK2* versus *Parkin*. This approach ensures that each final gene list represents a consistent transcriptional change linked to its respective experimental group. From these results, we compiled consistent, non-overlapping gene sets for each comparison. The biological functions of these distinct gene sets were subsequently investigated via ORA using WebGestalt (https://www.webgestalt.org) (Elizarraras et al, 2024). To account for potential confounders, we separately identified gender- and reprogramming-related effects. Using DESeq2, gender-associated DEGs were identified within the control group with the model ~ place + replicate + gender. Similarly, reprogramming-associated DEGs were identified within the *Parkin* group using the model ~ place + replicate + reprogramming_method.

### Functional characterization of DEGs via network and pathway analysis

To characterize the functional architecture of DEGs, we executed a stepwise PPI network analysis. First, a core PPI network was constructed from the DEGs using the STRING database v12.0 (Szklarczyk et al, 2023) with a mid- or high-confidence interaction score threshold of 0.4–0.7 with all active interaction sources. This core network was systematically expanded by incorporating first- and second-order interactors to reveal latent connectivity. The resulting network was partitioned into functional modules using the Markov cluster algorithm (MCL) (Van Dongen et al, 2008). Each module was subsequently refined by adding its own set of first- and second-order interactors. Finally, the biological relevance of the refined modules was assessed through functional pathway enrichment analysis using both the STRING database and WebGestalt. GSEA was applied to evaluate the statistically significant enrichment of these PPI-derived modules as custom gene sets with *P* value adjusted by Benjamini-Hochberg correction threshold (FDR) < 0.05. To ensure the identified clusters were not confounded by technical variables, we performed GSEA against ranked gene lists associated with gender and reprogramming methods. To complement our differential expression analysis and

comprehensively characterize enriched functional pathways, we conducted GSEA using clusterProfiler v4.12.6 (Wu et al, 2021). Genes were ranked by the signed metric $-\log_{10}(P\text{-value}) * \text{sign}(LFC)$. The analysis interrogated multiple canonical databases, including MSigDB (Castanza et al, 2023), Reactome (Milacic et al, 2024), GO (The Gene Ontology Consortium, 2021), and KEGG (Kanehisa et al, 2025), with gene set significance defined by a Benjamini-Hochberg FDR < 0.05. Cross-database gene identifier mapping was handled using the biomaRt package.

### Co-expression network analysis and integration with pathway activity

For co-expression analysis, we constructed a signed hybrid network using the BioNERO framework v1.12.0, leveraging both Pearson and Spearman correlation metrics for gene module identification. Correlations between module eigengenes were computed using the cor.test function in R, with FDR adjusted for multiple testing. To ensure data integrity, potential batch effects were preemptively corrected using the removeBatchEffect function from the limma package. The efficacy of this batch correction on differentiation site (place) variables was quantitatively confirmed by a significant reduction in the influence of batch factors, as assessed by PERMANOVA based on Euclidean distances of the VST data. Finally, to interpret the biological significance of the identified modules, we performed functional profiling through ORA using the STRING website or WebGestalt to evaluate custom gene sets derived from co-expression modules. Pathway activity analysis (PAA) scores were evaluated using progeny v1.30.0 (Schubert et al, 2018) and decoupleR v2.14.0 (Badia-i-Mompel et al, 2022) using batch-corrected transcription matrix. To evaluate the relationship between co-expression modules and pathway activity, we conducted a two-step analysis. First, PCA was performed on the four PD-associated co-expression modules to summarize their variation. Subsequently, the relationship between the first principal component (PC1) of each module and the predicted PAA scores was assessed by computing Spearman correlation coefficients. Statistical significance was determined using multiple correlation tests, with $P$ values adjusted by the FDR method. To assess the independent and combined contributions of biological and technical variables to pathway activity, we performed multiple linear regression. A generalized linear model (GLM) was fitted with PAA scores as the response variable, and mutation status, donor gender, reprogramming method, and laboratory of origin as explanatory variables, using the formula: PAA score ~ *mutation + gender + reprogramming + place*. All visualizations were generated using ggplot2 v3.5.1 supplemented by specialized packages including pheatmap v1.0.12, EnhancedVolcano v1.26.0, GseaVis v0.1.1, and UpSetR v1.4.0. The complete analytical workflow was implemented in R v4.4.2.

## Data Availability

Raw sequencing data from this study have been deposited in the NCBI SRA under BioProject ID PRJNA1346308. This dataset comprises 34 transcriptomes of iPSC-derived neuronal lines from PD patients and healthy controls, sequenced using Illumina NovaSeq 6000. Signatures for neuronal progenitors and mature neurons were defined based on the analysis of public RNA-seq data (GEO: GSE120271), which profiles cells across different stages of neuronal differentiation. All custom code used for transcriptomic processing and analysis has been deposited in the GitHub repository at https://github.com/cbio-lab/PARK2023.

## Supplementary Information

## Acknowledgements

This work was supported by the Ministry of Science and Higher Education of the Russian Federation (the Federal Scientific-technical programme for genetic technologies development for 2019–2030, agreement No 075-15-2025-518) (IV Kopylova, AN Bogomazova, MA Lagarkova, OS Lebedeva, EI Olekhnovich) and by the Grant No. 25-15-00165 (DA Grehnyov, VA Vigont) from the Russian Science Foundation.

### Author Contributions

IV Kopylova: investigation, visualization, methodology, and writing—original draft.
AB Ivanov: software, formal analysis, and investigation.
LR Eidelman: software.
EN Zaitseva: software.
ED Kulikova: software.
DA Grehnyov: investigation, methodology, and writing—original draft.
AN Bogomazova: resources, supervision, funding acquisition, and writing—review and editing.
VA Vigont: resources, funding acquisition, and writing—original draft, review, and editing.
EV Kaznacheyeva: resources, supervision, funding acquisition, and writing—review and editing.
MA Lagarkova: resources, supervision, funding acquisition, and writing—review and editing.
OS Lebedeva: conceptualization, supervision, project administration, and writing—original draft, review, and editing.
EI Olekhnovich: conceptualization, software, formal analysis, visualization, methodology, project administration, and writing—original draft, review, and editing.

### Conflict of Interest Statement

The authors declare that they have no conflict of interest.

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
