## [Reviewer comments · Life Science Alliance]

Convergent Transcriptomic Signature in iPSC-Dopaminergic Neurons of Hereditary Parkinson's Disease

Irina V. Kopylova, Artem B. Ivanov, Lev R. Eidelman, Ekaterina N. Zaitseva, Ekaterina D. Kulikova, Dmitriy A. Grehnyov, Alexandra Bogomazova, Vladimir A. Vigont, Elena V. Kaznacheyeva, Maria A. Lagarkova, Olga S. Lebedeva and Evgenii I. Olekhovich

DOI: <https://doi.org/10.26508/lsa.202503551>

Corresponding author(s): Dr. Evgenii I. Olekhovich (Federal Research and Clinical Center of Physical-Chemical Medicine named after Y.M. Lopukhin; Center for genetic reprogramming and gene therapy, Lopukhin Federal Research and Clinical Center of Physical-Chemical Medicine of Federal Medical Biological Agency)

Review Timeline:

Submission Date:	2025-10-28
Editorial Decision:	2025-12-04
Revision Received:	2026-02-06
Editorial Decision:	2026-03-17
Revision Received:	2026-03-25
Accepted:	2026-03-27

Scientific Editor: Tim Fessenden

Transaction Report:

December 4, 2025

Re: Life Science Alliance manuscript #LSA-2025-03551

Dr. Evgenii I. Olekhovich
Federal Research and Clinical Center of Physical-Chemical Medicine named after Y.M. Lopukhin
Pirogovskaya 1a
Moscow 123456
Russian Federation

Dear Dr. Olekhovich,

Thank you for submitting your manuscript entitled "Convergent Transcriptomic Signature in iPSC-Dopaminergic Neurons of Hereditary Parkinson's Disease" to Life Science Alliance. The manuscript was assessed by expert reviewers, whose comments are appended to this letter. We invite you to submit a revised manuscript addressing the Reviewer comments.

As you will see, reviewers appreciated the potential utility of this gene expression analysis for the field, but each expressed concerns that must be addressed in a revised manuscript. We concur with the major concerns of Reviewer 2 on the potential effects of the GBA1 variant, and on the FDR threshold selected both of which must be addressed. This Reviewer and Reviewer 1 both remarked on limitations inherent to this analysis that merit caution in drawing strong conclusions, and we agree the text should be adjusted according to their requests. Finally Reviewer 3 requested an improved analysis and presentation of eigengenes. We agree that the first two requests from Reviewer 3 would be helpful, however we leave these to your discretion. Similarly, although we agree that new experimental data sought by Reviewer 2 would strengthen the work, these are not required in a revision.

While you are revising your manuscript, please also attend to the below editorial points to help expedite the publication of your manuscript. Please direct any editorial questions to the journal office. When submitting the revision, please include a letter addressing the reviewers' comments point by point.

Thank you for this interesting contribution to Life Science Alliance. We hope that the comments below will prove constructive as your work progresses, and we are looking forward to receiving your revised manuscript.

Sincerely,

-- Summary blurb (enter in submission system): A short text summarizing in a single sentence the study (max. 200 characters including spaces). This text is used in conjunction with the titles of papers, hence should be informative and complementary to the title and running title. It should describe the context and significance of the findings for a general readership; it should be

written in the present tense and refer to the work in the third person. Author names should not be mentioned.

B. MANUSCRIPT ORGANIZATION AND FORMATTING:

Reviewer #1 (Comments to the Authors (Required)):

This manuscript reports the study of the transcriptome of iPSC-derived dopamine neurons, generated from individuals with genetic forms of Parkinson's disease and healthy controls. The dataset that has been generated and reported is interesting and is a useful advance for the field, and the data will be of value to the field for further work.

I would suggest that the conclusions drawn are rather strong given the evidence presented. Specifically, I think it is difficult to draw general conclusions about the direction of causality of the upregulated genomic instability pathways observed, not least given that this study includes samples from only two individuals for each of the LRRK2 and PARKIN mutations studied. Similarly, I am not sure to what extent it is reasonable to describe a "gradient" of pathology from the limited number of individuals studied.

Overall, I think this manuscript is suitable for publication but I would suggest changes particularly in the discussion to draw back from the suggestion of a model of pathogenesis (which cannot be true from this system; missing the contributions of glia, immune cells, α -synuclein etc.) and rather present a discussion of the divergent and convergent transcriptomic signatures observed.

Reviewer #2 (Comments to the Authors (Required)):

While this study could be of relevance to the field, presenting novel pathway affected in PD, several technical and experimental issues may substantially compromise the validity of the conclusions.

First, and most critically, the GBA1 mutation carried by the single patient harboring the LRRK2 G2019S variant is not addressed. Since this individual represents the only sample in the LRRK2 group, the potential confounding effects of the GBA1 mutation cannot be ignored.

Second, the false discovery rate (FDR) threshold of 25% is unusually permissive. This lack of statistical stringency, combined with the absence of orthogonal validation, is concerning and may undermine the robustness of the reported findings.

Although the authors candidly acknowledge some limitations in the Discussion, additional validation experiments would have been feasible and would have strengthened the transcriptomic conclusions.

Major comments

- Reprogramming method: Fibroblasts were reprogrammed using either lentiviral or Sendai virus approaches. Please clarify the rationale for using different methods and provide evidence that this does not introduce technical or biological bias.
- Principal component analysis (PCA): The PCA shown in Figure S2 suggests a stronger clustering by batch or site ("place") than by genotype. In addition, the number of plotted points does not correspond to the stated 34 transcriptomic samples. Moreover, all PD donors were male whereas one control donor was female. These variables are likely to have a substantial impact on the results.
- Heatmaps (Figure 3): The heatmaps appear to be influenced by technical or site-related effects rather than genotype, reinforcing concerns about batch confounding.
- LRRK2-GBA1 genetic interaction: The interaction between LRRK2 and GBA1 variants is an active and unresolved area of research in Parkinson's disease. Patients carrying both mutations often show distinct phenotypes, potentially due to a protective interaction between LRRK2 kinase activity and GCase function. Ignoring this interaction in a donor harboring mutations in both genes is highly problematic and likely affects all downstream analyses. Inclusion of GBA1-only and LRRK2-only donors would have been necessary.
- Sample size and independence: The number of independent samples (n) is unclear throughout the manuscript. Please clarify what is considered an independent biological replicate (donor vs. derived cell line). It is also unclear why the PCA includes fewer

data points than expected based on the reported sample number.

- Statistical thresholds: The use of a Benjamini-Hochberg adjusted p-value threshold of < 0.25 is highly unconventional. The authors should justify this choice. In Figure 2 (volcano plots), the extreme $-\log_{10}(p)$ values appear to be driven by a small subset of genes; these genes should be explicitly identified.
- Pathway analysis thresholds: An FDR threshold of < 0.25 for pathway enrichment analysis (Figure 2) is also excessively permissive and raises concerns about false-positive findings.
- Supplementary Table 4 (ORA analysis): The number of overlapping genes supporting each enriched pathway is very small. If these represent the intersection of the gene list and pathway set, a minimum gene-count threshold should be applied to ensure biological relevance.
- Figure 3B variability: The variability appears very high. Although the authors describe a trend, the adjusted p-values for both the Parkin and LRRK2 groups exceed conventional significance thresholds.
- TRAIL pathway: One of the main conclusions concerns dysregulation of the TRAIL pathway, yet this is not shown clearly in any of the main figures.
- Speculative discussion: Several mechanistic interpretations (cell junctions, DNA damage, genome stability, potassium channels, apoptosis) are largely speculative. Many of these could have been directly tested in the presented cellular model.
- Overgeneralization: The manuscript occasionally generalizes findings to Parkinson's disease broadly. However, the study focuses only on two genetic forms of PD, and such generalizations should be made with caution.

Minor comments

- Define all abbreviations at first use.
- The Parkin mutations assessed are heterogeneous. Please clarify their functional effects and whether they uniformly reduce protein expression. Most mutations appear to be heterozygous and are not associated with increased PD risk in the general population. Please justify this choice.
- Please justify the selection of the 52-day differentiation time point.
- TH immunostaining in Figure 1 is not convincing, and the cultures appear to contain substantial debris.
- Marker gene expression should be compared to undifferentiated iPSCs and/or other neuronal subtypes to demonstrate dopaminergic enrichment.
- There are errors in supplementary figure citations (e.g., variance explained is in Figure S2B rather than S3).
- Figure 2 is difficult to interpret due to poor readability.
- Does the line PDP4.4S (Park14 cl4) carry a mutation in the Park14 gene? Please clarify.
- The authors report increased translational activity in Parkin PD models. Please comment on mitochondrial translation specifically.
- The Discussion mentions mtDNA in Parkin PD. Recent studies have also implicated LRRK2 in lysosomal mtDNA release and pyroptosis; this should be acknowledged.

Reviewer #3 (Comments to the Authors (Required)):

Kopylova et al. reported transcriptomic analyses on Parkinson's disease (PD) patient derived dopaminergic neurons (DANs) differentiated from induced pluripotent stem cell (iPSC). Focusing on pathological mutations, authors sequenced 34 differentiated DANs from one PD patient bearing LRRK2 (p.G2019S) & GBA1 (p.N370S) mutations, three patients bearing various PRKN mutations, and two healthy controls, with appropriate biological and technical replicates. With Gene Set Enrichment Analysis (GSEA) and co-expression network analysis on differentially expressed genes (DEGs), they identified epigenetic dysregulation and TGF- β related biological functions correlated with the hypothesized disease severity axis of healthy, LRRK2 mutation and PRKN mutation. The mutation specific co-expression networks were also reported.

Overall, the structure of the manuscript is clear and straightforward, and the transcriptomics data will be useful for the field in studying the molecular pathogenesis of related mutations. However, improvements can be made to increase data transparency and interpretability.

Major points:

1) In Figure 1 the expression of dopaminergic markers in three groups were shown. It demonstrated the homogeneity of DANs across three groups, but the abundance of DANs relative to other cell types can be hard to interpret based on the values. This could be improved by including markers of other cell types in the plot, such as stem cell markers (SOX2, POU5F1, NANOG) and glial cell markers (GFAP, OLIG2). In addition, the cell cycle analysis could also be supported by the expression of typical proliferation markers (MKI67, TOP2A).

2) The authors described cell culture place as a major source of variation in the transcriptomics. It is a known and debated issue in iPSC differentiated neurons, as highlighted in several studies:

- Volpato, Viola, et al. "Reproducibility of molecular phenotypes after long-term differentiation to human iPSC-derived neurons: a multi-site omics study." *Stem cell reports* 11.4 (2018): 897-911.
- Strano, Alessio, et al. "Variable outcomes in neural differentiation of human PSCs arise from intrinsic differences in developmental signaling pathways." *Cell reports* 31.10 (2020).
- Burke, Emily E., et al. "Dissecting transcriptomic signatures of neuronal differentiation and maturation using iPSCs." *Nature communications* 11.1 (2020): 462.

- Reed, Xylena, et al. "Transcriptional signatures in iPSC-derived neurons are reproducible across labs when differentiation protocols are closely matched." *Stem cell research* 56 (2021): 102558.
- Beekhuis-Hoekstra, Stephanie D., et al. "Systematic assessment of variability in the proteome of iPSC derivatives." *Stem Cell Research* 56 (2021): 102512.
- Jerber, Julie, et al. "Population-scale single-cell RNA-seq profiling across dopaminergic neuron differentiation." *Nature genetics* 53.3 (2021): 304-312.

Although the scope of the manuscript is to examine the mutation related biological effect, the authors are in a unique position to evaluate such differences, as they performed parallel differentiation with the same cell lines at different laboratories. I suggest that authors report the related results in supplementary information, for instance, the DEGs by comparing two places, and the genes in corresponding Pearson correlation co-expression modules (MEdarkturquoise, MEviolet, MEorangered4, MElightcyan1). These results could then be summarized succinctly with three sentences in the main manuscript.

3) The comparison of eigengenes from identified disease related modules can provide a global view of the gene changes within those modules. However, understanding module homogeneity and effect size of the changes requires examination on the expression of each gene. A common way to facilitate this is by visualizing gene expression in the same module with heatmaps. While heatmaps are not necessarily for inclusion in the manuscript, they may provide insights into the driver genes in the module, which can be co-examined with protein-protein interaction (PPI) networks. Practically, the genes can be ranked by fold change, and genes with the largest changes can be labeled in PPI networks in the Figures. Conversely, nodes (genes) in a PPI network can be ranked by simple metrics such as degree centrality, and the expression change of top ranked nodes can be plotted. While it is the authors' preference which method to choose, narrowing down the number of targets in a module using certain criteria can help biological interpretation and add more depth to the global view. In addition, I suggest that authors describe the definition of the edges in the PPI network in Methods or figure legends.

Minor points:

- 1) Discussion paragraph 8 has a long description of the proposed pathogenesis model based on module functions. It may have more translational value if authors can shorten this part by proposing several actionable biological experiments that can facilitate hypothesis validation.
- 2) Results-Data overview last paragraph, Figure S3 should likely be Figure S2B
- 3) Figure 2 legend, (D) Healthy control-associated, and (E) LRRK2-/Parkin-associated should likely be (D) LRRK2 unique, and (E) Parkin unique
- 4) AAO was used first time in Introduction paragraph 3, please put the full name.
- 5) Materials and methods- Molecular Biology Reagents- TRIzol country code CUSA, is it a typo?
- 6) iPSC, IPSC should likely be iPSC; TGF-beta, TGF- β should likely be TGF- β

Dear Editor and Reviewers,

Thank you for the opportunity to submit a revised version of our manuscript. We are grateful to the reviewers for their insightful and constructive feedback, which has been invaluable in strengthening our work. We have carefully addressed all comments and suggestions. All changes in the manuscript have been highlighted in the attached file 'changes.docx'. The revisions have significantly improved the manuscript, and we have provided a detailed point-by-point response to each reviewer comment below. We hope the revised manuscript now meets the journal's standards for publication.

Sincerely,
on behalf of all authors,
Evgenii Olekhovich

Reviewer #1 (Comments to the Authors (Required)):

This manuscript reports the study of the transcriptome of iPSC-derived dopamine neurons, generated from individuals with genetic forms of Parkinson's disease and healthy controls. The dataset that has been generated and reported is interesting and is a useful advance for the field, and the data will be of value to the field for further work.

I would suggest that the conclusions drawn are rather strong given the evidence presented. Specifically, I think it is difficult to draw general conclusions about the direction of causality of the upregulated genomic instability pathways observed, not least given that this study includes samples from only two individuals for each of the LRRK2 and PARKIN mutations studied. Similarly, I am not sure to what extent it is reasonable to describe a "gradient" of pathology from the limited number of individuals studied.

Overall, I think this manuscript is suitable for publication but I would suggest changes particularly in the discussion to draw back from the suggestion of a model of pathogenesis (which cannot be true from this system; missing the contributions of glia, immune cells, a-synuclein etc.) and rather present a discussion of the divergent and convergent transcriptomic signatures observed.

We sincerely thank the Reviewer for their careful reading of our manuscript and their constructive feedback. We agree that our initial interpretations, particularly in the Discussion section, were overly strong and extended beyond the scope of what can be concluded from our data. We have revised the manuscript accordingly to temper our conclusions and more accurately reflect the correlative nature of our transcriptomic findings. In line with your comments, we have significantly reframed the narrative. We now present the observed molecular patterns as a "convergent signature" evident in our dataset, explicitly stating that this represents a correlative observation that requires validation in larger cohorts. We have removed the definitive language suggesting a proven "gradient of pathology" or an "organizing principle" of pathogenesis. The Discussion has been rewritten to focus on a comparative analysis of the convergent and divergent transcriptomic signatures, which was the core aim of our study, rather than presenting an overarching model of pathogenesis. We have toned down the speculative mechanistic cascade and removed detailed hypotheses about non-cell-autonomous factors not present in our monoculture system. We have also strengthened the Limitations section to explicitly state that our iPSC-derived neuronal monoculture cannot model the contributions of glia, immune cells, or α -synuclein pathology, and that all mechanistic inferences require direct experimental validation. Changes are marked in the 'changes.docx' file on pages 42-52, lines 1296-1660.

Reviewer #2 (Comments to the Authors (Required)):

While this study could be of relevance to the field, presenting novel pathways affected in PD, several technical and experimental issues may substantially compromise the validity of the conclusions. First, and most critically, the GBA1 mutation carried by the single patient harboring the LRRK2 G2019S variant is not addressed. Since this individual represents the only sample in the LRRK2 group, the potential confounding effects of the GBA1 mutation cannot be ignored. Second, the false discovery rate (FDR) threshold of 25% is unusually permissive. This lack of statistical stringency, combined with the absence of orthogonal validation, is concerning and may undermine the robustness of the reported findings. Although the authors candidly acknowledge some limitations in the Discussion, additional validation experiments would have been feasible and would have strengthened the transcriptomic conclusions.

Major comments

1) Reprogramming method: Fibroblasts were reprogrammed using either lentiviral or Sendai virus approaches. Please clarify the rationale for using different methods and provide evidence that this does not introduce technical or biological bias.

We are grateful to the Reviewer for raising this essential methodological point regarding the use of different reprogramming methods. We fully recognize that the use of different reprogramming methods (lentivirus and Sendai virus) could potentially be a source of technical or biological variation [Churko et al., 2017]. To rigorously address this concern and isolate the specific biological signature of the disease, we implemented a multi-layered analytical strategy. First, we employed a convergent approach where the core PD common differential expression signature was derived from the intersection of results from various comparisons, including those involving only *LRRK2* samples (all reprogrammed exclusively using Sendai virus). Since these specific comparisons contained no lentivirus-reprogrammed samples, any hypothetical transcriptomic signal specific to the lentiviral method could not manifest in these DEG lists. Second, and critically important, we performed a dedicated analysis to identify DEGs potentially associated with reprogramming methodology using DESeq2 with an appropriate statistical model. This yielded 1,982 reprogramming-associated DEGs (1,034 upregulated, 948 downregulated) that we used for comprehensive GSEA analyses. We systematically tested the enrichment of these method-specific signatures against all our experimental group comparisons and, most significantly, against our key PPI-derived functional clusters. This allowed us to quantitatively assess whether our identified disease signatures co-varied with reprogramming artifacts. Our GSEA validation revealed that while certain pathways (particularly ECM organization and epigenetic regulation) showed significant association with reprogramming methods, our core convergent PD signature — characterized by coordinated downregulation of WNT/ β -catenin signaling and cell cycle pathways, along with upregulation of neuronal specialization pathways as voltage-gated potassium channels or transmission across chemical synapses — remained robust and showed no significant enrichment for reprogramming-associated genes. At the same time, we specifically note in the Discussion that some unique transcriptomic effects observed in the Parkin mutation group may be more susceptible to potential confounding influence from the reprogramming method. Therefore, we interpret these findings with particular caution, labeling them as preliminary and requiring further independent validation in studies with uniform reprogramming methodology. We acknowledge the inherent limitations of heterogeneous reprogramming approaches and have explicitly stated this as a methodological limitation in our paper, while demonstrating through rigorous bioinformatic controls that our central conclusions remain robust to these

technical complications. A detailed analysis of the effects of reprogramming is provided in the document 'changes.docx' on pages 32-33, lines 964-1005.

1) Churko, Jared M., et al. "Transcriptomic and epigenomic differences in human induced pluripotent stem cells generated from six reprogramming methods." *Nature Biomedical Engineering* 1.10 (2017): 826-837.

2) Principal component analysis (PCA): The PCA shown in Figure S2 suggests a stronger clustering by batch or site ("place") than by genotype. In addition, the number of plotted points does not correspond to the stated 34 transcriptomic samples. Moreover, all PD donors were male whereas one control donor was female. These variables are likely to have a substantial impact on the results.

We thank the Reviewer for this careful observation regarding the PCA plot. We acknowledge that the PCA plot in Figure S2, in its current form, is suboptimal for visual interpretation. The seemingly reduced number of data points is, in part, a graphical artifact. This occurred because the high transcriptomic similarity between specific biological replicates, such as different iPSC clones from the same donor, caused their data points to overlap almost completely on the 2D plot, which obscures the true sample count and can visually exaggerate technical dispersion. The analysis was indeed performed on all 34 transcriptomic samples. We fully agree that the site of sample processing is a major technical variable and proactively addressed this confounder. In the differential expression analysis using DESeq2, the place variable was explicitly included as a covariate in the statistical model. Consequently, while the uncorrected data in this PCA may show site-related structure, the core biological results of our study are derived from data where this technical effect has been statistically accounted for. We agree the figure is misleading and revised Figure S2 for the final manuscript. The updated version used transparent symbols to prevent point overlap, ensuring all samples are visible.

We sincerely thank the Reviewer for raising this critical point concerning the influence of biological gender on transcriptomic profiles. We acknowledge that biological gender can significantly influence transcriptomic profiles and disease mechanisms. To rigorously account for this variable and ensure our findings reflect disease-specific biology rather than gender-related confounding effects, we implemented a multi-tiered analytical framework analogous to our approach for reprogramming methodology. First, to generate overlapping sets of genes common to both mutations and to differentiate these mutations from both controls and each other, in addition to comparisons with control samples, we made comparisons between LRRK2 vs Parkin groups that consisted of male samples only.

Second, we performed a dedicated gender-stratified analysis by identifying gender-associated differentially expressed genes (DEGs) in control iPSC-derived dopaminergic neurons. Using GSEA, we systematically evaluated whether our core disease signatures — including convergent downregulation of WNT/ β -catenin signaling and cell cycle pathways, and upregulation of pathways included to neuronal specialization — showed significant overlap with gender-associated transcriptional programs. Critically, none of these core pathological signatures demonstrated enrichment for gender-linked DEGs (all FDR > 0.05), confirming their independence from gender-related biological variation. A detailed analysis of comparison male and female control cell lines is provided in the document 'changes.docx' on page 32, lines 946-962.

3) Heatmaps (Figure 3): The heatmaps appear to be influenced by technical or site-related effects rather than genotype, reinforcing concerns about batch confounding.

We sincerely thank the Reviewer for this critical insight, which addresses a fundamental question regarding the integrity of our co-expression analysis. We understand the concern that the observed patterns in the co-expression modules might reflect unresolved technical variation. However, our bioinformatic pipeline was explicitly designed to isolate biological signals from technical confounders. Prior to co-expression analysis, we applied a validated batch correction method to remove variance attributable to the differentiation site ("place" variable). The efficacy of this correction was quantitatively confirmed by PERMANOVA, which showed that the contribution of the "place" factor to the overall transcriptomic variance became negligible after correction ($R^2 = 0.01$, $p = 0.98$), while the variance explained by the disease condition remained highly significant. The subsequent identification of location-associated modules does not indicate a failure of batch correction, but rather reflects the successful retention of meaningful biological heterogeneity. These modules are enriched for pathways governing neuronal maturation and developmental timing. This suggests that differentiations performed in the two laboratories yielded populations at slightly distinct points along a neurodevelopmental continuum — a biologically plausible outcome even with standardized protocols. Crucially, the PD-associated modules (e.g., skyblue3, black) were distinctly separate from these location-associated modules. They were specifically and significantly correlated with disease status and enriched for pathways related to axonogenesis and synaptic structure, which aligns with our core pathological model. In summary, the batch correction successfully mitigated technical noise, allowing us to distinguish between two layers of biological signal: one related to inherent differences in

developmental pacing across laboratories, and a separate, robust signature directly associated with the genetic etiology of PD.

4) LRRK2-GBA1 genetic interaction: The interaction between LRRK2 and GBA1 variants is an active and unresolved area of research in Parkinson's disease. Patients carrying both mutations often show distinct phenotypes, potentially due to a protective interaction between LRRK2 kinase activity and GCase function. Ignoring this interaction in a donor harboring mutations in both genes is highly problematic and likely affects all downstream analyses. Inclusion of GBA1-only and LRRK2-only donors would have been necessary.

We sincerely thank the Reviewer for this critical and insightful comment, which rightly highlights one of the most significant methodological challenges in our study. We fully acknowledge that the interaction between *LRRK2* and *GBA1* variants is a complex and active area of research. The presence of the co-occurring *GBA1* N370S mutation in our single *LRRK2* G2019S donor line is a major genetic confounder, and we have addressed this fundamental limitation transparently in the second paragraph of our "Limitations" section. We agree that an ideal design would require matched cohorts with isolated mutations and isogenic lines to disentangle individual and synergistic effects.

Our interpretation and analytical strategy were designed with this specific caveat in mind. The central aim of this study was to identify a convergent pathological signature across genetically distinct forms of PD, not to define a pure *LRRK2* molecular signature. Therefore, the most critical and robust findings are the transcriptomic disruptions shared between this complex *LRRK2/GBA1* line and the Parkin-mutant lines. We argue that these overlapping pathways — namely the suppressed pathways which we define as involved in maintaining cell cycle and WNT/ β -catenin signaling alongside upregulated pathways included to neuronal specific functions — are more likely to represent a core, disease-relevant pathology than an artifact specific to the *GBA1* variant. When interpreting the *LRRK2*-specific profile, we operated under the assumption, noted in our limitations, that the phenotype was predominantly driven by the *LRRK2* G2019S mutation. This is supported by literature indicating the *GBA1* N370S variant has relatively low penetrance and that in clinical cohorts with both variants, the disease phenotype is primarily associated with the *LRRK2* genotype [Anheim et al., 2012; Zhang et al., 2018; Ortega et al., 2021]. We explicitly stated that we cannot exclude a modifying role for the *GBA1* variant. In direct response to this comment, we have amended the manuscript to more clearly frame our findings. In the Limitations, we more emphatically stated that the unique *LRRK2*-associated signature should be considered preliminary and requires validation in models without confounding

GBA1 mutations. We thank the Reviewer for this valuable critique, which allows us to present a more nuanced interpretation of our data and a clearer roadmap for subsequent research. These changes are highlighted in the 'changes.docx' file on page 45, lines 1414-1429.

References

- 1) Ortega, Roberto A., et al. "Association of dual LRRK2 G2019S and GBA variations with Parkinson disease progression." *JAMA Network Open* 4.4 (2021): e215845-e215845.
- 2) Zhang, Li-Ying, et al. "Role of histone deacetylases and their inhibitors in neurological diseases." *Pharmacological Research* 208 (2024): 107410.
- 3) Anheim, M., et al. "Penetrance of Parkinson disease in glucocerebrosidase gene mutation carriers." *Neurology* 78.6 (2012): 417-420.

5) Sample size and independence: The number of independent samples (n) is unclear throughout the manuscript. Please clarify what is considered an independent biological replicate (donor vs. derived cell line). It is also unclear why the PCA includes fewer data points than expected based on the reported sample number.

We thank the Reviewer for raising these crucial methodological points regarding sample size and the definition of biological replicates. We clarified these aspects in the revised manuscript. In the context of our study, the donor is considered an independent biological replicate. The analysis included transcriptomes from neurons derived from 6 independent donors: 2 healthy controls, 1 *LRRK2/GBA1* carrier, and 3 *Parkin* carriers. Therefore, biological variation is represented by the differences between these individuals within each condition (disease status/mutation). The total of 34 analyzed transcriptomic samples originated from these 6 donors through a multi-step process designed to assess experimental robustness. Each iPSC line was differentiated into mature dopaminergic neurons. This differentiation was performed independently in two laboratories. Within each differentiation experiment at each laboratory, we generated two technical replicates — independent RNA isolations and library preparations from the same pool of mature neurons — to capture variation in the final processing steps. Consequently, most donors yielded multiple samples (2 labs × 2 technical replicates = 4 samples), summing to 34 samples in total as detailed in Table 1. All 34 samples were included in the bioinformatic analysis. We acknowledge that the initial visualization in the PCA plot (Figure S2) was suboptimal. The apparent reduction in visible data points was a graphical artifact caused by the near-complete transcriptomic overlap of technical replicates from the same donor, which led to their points superimposing on the 2D plot. This inadvertently obscured the true sample

count. It is important to note that the PCA analysis was in fact performed using all 34 samples. We corrected Figure S2 in the final manuscript by using transparent plotting symbols to ensure all samples are visible and to prevent any misinterpretation.

6) Statistical thresholds: The use of a Benjamini-Hochberg adjusted p-value threshold of < 0.25 is highly unconventional. The authors should justify this choice. In Figure 2 (volcano plots), the extreme $-\log_{10}(p)$ values appear to be driven by a small subset of genes; these genes should be explicitly identified.

We sincerely thank the Reviewer for highlighting this important statistical consideration. We appreciate the opportunity to clarify our methodological rationale and agree that transparency regarding the application of significance thresholds is essential. You are correct that a FDR threshold of < 0.25 is unconventional for declaring significance in a final analysis. We would like to clarify its specific and limited application in our study. This relaxed threshold (FDR < 0.25) was used exclusively in specific exploratory steps. Its primary application was in the initial ORA performed on the consistent sets of DEGs to generate biological hypotheses, as noted in the manuscript. All primary results and definitive conclusions in our study are based on stringent downstream methods. The functional pathways and mechanisms discussed are supported by robust results from protein-protein interaction network analysis and systematic GSEA, where all reported enrichments meet the conventional significance threshold of FDR < 0.05. Consequently, none of the conclusions presented rely on findings generated using the adj. $p < 0.25$ cutoff. Therefore, our core findings do not rely on any results from the initial exploratory screen. The use of the relaxed cutoff was confined to early data interrogation, while the substantive outcomes of the paper rested on analyses performed with standard statistical rigor.

Regarding the genes with extreme $-\log_{10}(p)$ values in the volcano plots, we agree that a small subset of genes can drive such patterns. While these individual genes are statistically significant, our analytical strategy intentionally does not interpret them in isolation. We explicitly identify these genes in the underlying data files accompanying the manuscript. However, the core of our biological interpretation relies on methods that operate on coordinated gene sets, such as the STRING networks or functional pathway analyses. This approach is predicated on the principle that while a single gene with extreme differential expression could potentially be an analytical artifact or outlier, the coordinated behavior of an entire functional pathway is far less likely to arise by random chance. This strategy provides a more robust and biologically contextualized framework for identifying key dysregulated processes in Parkinson's disease, moving beyond a focus on individual gene candidates which may be more susceptible to technical noise or batch effects.

7) Pathway analysis thresholds: An FDR threshold of < 0.25 for pathway enrichment analysis (Figure 2) is also excessively permissive and raises concerns about false-positive findings.

We thank the Reviewer for raising this important point regarding the statistical thresholds used in our pathway analysis. We acknowledge that an FDR < 0.25 is an unusually permissive threshold for a standard, standalone pathway enrichment analysis. We would like to clarify that the analysis presented in Figure 4 (former Figure 2) represents a specific, multi-step workflow based on a convergent evidence philosophy, rather than a single, conventional pathway enrichment test. In this workflow, we first identify a robust set of DEGs from consistent intersections. We then use an initial ORA to characterize these gene lists. At this exploratory stage, we intentionally apply more lenient thresholds. The rationale is that biological signals which are even weakly suggested at this stage, if they are subsequently recapitulated and validated through independent, orthogonal methods, provide stronger convergent evidence than a single stringent test. Critically, this initial ORA was not the final step. We used the core DEGs to construct PPI networks via STRING, expanding them to uncover latent functional diversity and define functional modules. The biological themes of these computationally derived PPI modules consistently aligned with the pathways suggested by the initial ORA, providing the first layer of validation. Most importantly, to definitively rule out noise and provide robust statistical evidence, we used these expanded PPI modules as custom gene sets in a GSEA. The results shown in Figure 4E (former Figure 2E) report the GSEA statistics, for which all significant modules have an adjusted p-value (FDR) < 0.05 . Therefore, the final, reported conclusions regarding enriched biological pathways are supported by this stringent statistical threshold within the GSEA framework. We have added the GSEA results to Supplementary Table S14 of the current version of the article.

8) Supplementary Table 4 (ORA analysis): The number of overlapping genes supporting each enriched pathway is very small. If these represent the intersection of the gene list and pathway set, a minimum gene-count threshold should be applied to ensure biological relevance.

We sincerely thank the Reviewer for this astute methodological observation regarding the gene overlaps in our initial ORA results in Supplementary Table S11 (former Supplementary Table S4). We acknowledge that for a standard, standalone over-representation analysis, small gene set intersections can be a valid concern and often warrant the application of a

minimum-count threshold to ensure biological relevance. We would like to clarify that the role of this initial ORA within our specific analytical framework was exploratory rather than declarative. In our study, this ORA step served as the first stage in a convergent evidence pipeline. Its purpose was to generate initial functional hypotheses from our core set of differentially expressed genes using more lenient thresholds. These pathway hints, even those supported by modest gene overlaps, were not considered final results. They subsequently served as input for independent and more rigorous validation steps. The key biological themes suggested by the ORA were investigated by using the core gene lists to construct expanded protein-protein interaction networks. This network approach was specifically chosen to move beyond the initial gene counts and to define broader, coherent functional modules based on protein interactions. The ultimate validation of biological relevance was then provided by using these computationally derived network modules as custom gene sets in a stringent GSEA. The statistical significance reported for these modules in our final analysis (Figure 4E - former Figure 2E, Supplementary Table S14) relies on a conventional GSEA FDR threshold of < 0.05 . Therefore, the relevance of the pathways discussed is substantiated not by the size of the initial ORA overlap, but by their convergent recapitulation and validation through downstream network analysis and statistically robust GSEA.

9) Figure 3B variability: The variability appears very high. Although the authors describe a trend, the adjusted p-values for both the Parkin and LRRK2 groups exceed conventional significance thresholds.

We sincerely thank the Reviewer for this important observation regarding the variability in Figure 3B and the associated statistics.. Your point is well taken. You are correct that the individual comparisons of the LRRK2 group versus controls and the Parkin group versus controls do not meet the conventional threshold for adjusted statistical significance when considered in isolation. This level of variability initially led us to also report and interpret the directional trend observed in these individual comparisons. However, to avoid any potential for misinterpretation and to uphold the most stringent standards of reporting, we have removed this specific analysis and interpretation of trends from the revised manuscript. Our primary analytical approach, driven by the study's central hypothesis, was to test for a convergent pathological signature. Therefore, we combined the LRRK2 and Parkin cohorts into a single PD group for comparison against the healthy controls. This analysis yielded a statistically significant result. We interpret this as evidence that while individual mutations introduce noise and unique secondary effects, pooling the groups allows the shared transcriptional "signal" of a convergent pathology to be detected above the background

variability. This finding is not an isolated statistical outcome but is biologically coherent. It aligns directly with other core results in the manuscript that identify a convergent transcriptomic signature — such as the downregulation of WNT/ β -catenin signaling and upregulation of neuronal specialization pathways — shared by both LRRK2 and Parkin models. The statistical significance in the combined cohort, therefore, serves as a valid test supporting our model of a common pathological trajectory in hereditary PD.

10) TRAIL pathway: One of the main conclusions concerns dysregulation of the TRAIL pathway, yet this is not shown clearly in any of the main figures.

We sincerely thank the Reviewer for underscoring the significance of the TRAIL-mediated apoptotic pathway dysregulation. In direct response to this valuable critique, we have created a new Figure 5 in the revised manuscript to present these results prominently within the main narrative. We acknowledge that our initial, more cautious placement in the supplementary materials may have undersold the finding's potential importance. Our standard approach is to prioritize for the main figures those signatures with the broadest support across multiple analytical methods. While the TRAIL pathway activity was robustly identified as upregulated in both LRRK2 and Parkin neurons through our pathway activity analysis, we now agree with the Reviewer that its coherent alignment with models of potentially aberrant neuronal cell structure in PD merits a more central presentation.

11) Speculative discussion: Several mechanistic interpretations (cell junctions, DNA damage, genome stability, potassium channels, apoptosis) are largely speculative. Many of these could have been directly tested in the presented cellular model.

We agree with the Reviewer that several mechanistic interpretations in the original discussion were overly speculative. In response, we have significantly revised the Discussion section to reduce its speculative nature. We have refocused the narrative on describing the key transcriptional phenomena observed in our analysis — such as the convergent regulation of specific pathways and the distinct profiles associated with different genetic backgrounds. The revised discussion now more conservatively links our core findings to the existing literature and highlights their potential implications. We believe this edit has strengthened the manuscript by ensuring a more precise and data-driven interpretation. These changes are highlighted in the 'changes.docx' file on pages 42-45, lines 1296-1405.

12) Overgeneralization: The manuscript occasionally generalizes findings to Parkinson's disease broadly. However, the study focuses only on two genetic forms of PD, and such generalizations should be made with caution.

We are grateful to the Reviewer for highlighting the need for caution in generalizing our findings. We agree that overgeneralization from specific genetic forms to the broader Parkinson's disease population should be avoided. In response to your comment, we have carefully revised the Discussion section of the manuscript to significantly reduce such broad claims. This ensures our conclusions are appropriately cautious and data-driven. These changes are highlighted in the 'changes.docx' file on pages 42-45, lines 1296-1405.

Minor comments

1) Define all abbreviations at first use.

Thank you for this fair point. We will revise the text for the final manuscript.

2) The Parkin mutations assessed are heterogeneous. Please clarify their functional effects and whether they uniformly reduce protein expression. Most mutations appear to be heterozygous and are not associated with increased PD risk in the general population. Please justify this choice.

We would like to thank the Reviewer for their valuable observation. Although three *Parkin* patients presented with an early-onset form of Parkinson's disease (PD), they actually had heterogeneous mutations in the *Parkin* gene.

One patient had complex heterozygous mutations, including a deletion of 202-203 AG in the second exon and a splicing mutation in the first intron (IVS1+1G/A). Compound heterozygous mutations usually lead to the loss of function of Parkin protein and cause an autosomal recessive form of PD with an early onset. While these mutations often result in a significant decrease in Parkin protein activity, they may not always lead to a permanent decrease in protein expression across all tissues. There have been reports of a complete loss of Parkin protein in the primary skin fibroblasts of patients with heterozygous mutations, regardless of the specific type of mutation [Zilocchi et al., 2020]. In addition, even when the mutation does not result in a substantial decrease in the amount of protein produced, it can still disrupt its enzymatic function, making "decreased expression" an unreliable indicator of the severity of the condition.

While homozygous or compound heterozygous mutations can cause autosomal

recessive early-onset PD, the effects of a single heterozygous deletion or duplication of the second exon of the *Parkin* gene, as presented in two other patients, are controversial. Large-scale studies, including data from the UK Biobank, suggest that such heterozygous carriers are just as likely to develop PD as non-carriers, indicating that it is often a benign variation in the general population [Zhu et al., 2022]. However, the same study also showed that single mutations in *Parkin* can still have a subclinical effect on dopaminergic neurons, leading to a slight dopamine deficiency in the striatum compared to controls. Additionally, heterozygous copy number variations causing deletions or duplications of exons in *Parkin* transcript sometimes do associate with early-onset PD [Petrucci et al., 2017]. At the same time, the expression of Parkin protein in heterozygous carriers is not uniformly reduced, and it varies significantly.

Taken together, we assume that the two *Parkin* patients with heterozygous mutations having been included in this study were genetically predisposed to PD, and other factors could also contribute to the pathogenesis. Since all *Parkin* patients included in the analysis had an early-onset of PD, our research focuses on the similarities in their transcriptomic profiles.

- 1) Zilocchi, Mara, et al. "Exploring the impact of PARK2 mutations on the total and mitochondrial proteome of human skin fibroblasts." *Frontiers in cell and developmental biology* 8 (2020): 423.
- 2) Zhu, William, et al. "Heterozygous PRKN mutations are common but do not increase the risk of Parkinson's disease." *Brain* 145.6 (2022): 2077-2091.
- 3) Petrucci, Simona, et al. "Genetic paradoxes in an Italian family with PARK2 multiexon duplication." *Movement Disorders Clinical Practice* 4.6 (2017): 889.

3) Please justify the selection of the 52-day differentiation time point.

We appreciate the Reviewer's request for clarification on our experimental methodology. We differentiated iPSCs into dopaminergic neurons as it was described in our previous work Lebedeva et al. (2023). The protocol consists of three main stages. First, neuronal differentiation is induced by dual inhibition of the SMAD signaling pathway, leading to generation of early neural progenitors (also known as neuroepithelial stem cells or progenitor cells). The next stage provides the ventral midbrain paternalization, which is followed by the stage of neuronal maturation. The first stage of differentiation lasts 2 weeks, the second stage lasts 10 days. The duration of the last stage can vary, but it should be at least 2 weeks long. Throughout the development of this protocol, we had assessed the quality of the cells

and had concluded that they could be used for experiments starting from day 38 of differentiation.

However, when working with dopaminergic neurons derived from patient-specific iPSCs, we extend the duration of the last stage to 4 weeks in order to ensure full maturation and increase the possibility of phenotype manifestation. Previously we had evaluated the maturity of dopaminergic neurons and their functional activity determined by spontaneous dopamine release and electrical excitability on day 52. Taken together, the 52-day time point of differentiation was chosen for this work.

1) Lebedeva, Olga S., et al. "An Efficient 2D Protocol for Differentiation of iPSCs into Mature Postmitotic Dopaminergic Neurons: Application for Modeling Parkinson's Disease." *International Journal of Molecular Sciences* 24.8 (2023): 7297.

4) TH immunostaining in Figure 1 is not convincing, and the cultures appear to contain substantial debris.

Figure A1. Representative flow cytometry analysis of DANs on day 52 of differentiation. The data are presented as overlaid histogram plots of anti-TH antibody-stained cells (brown) and isotype control antibody-stained cells (blue).

We thank the Reviewer for raising this important point regarding the characterization of our cultures. We fully agree that a single immunostaining image is insufficient for robust quantification, and we appreciate the opportunity to clarify our validation approach. The neuronal cultures in this study were generated using our established, highly efficient differentiation protocol (Lebedeva et al., 2023), which we have previously validated for purity and reproducibility. For the specific iPSC line PDP1.5L (differentiated in Moscow), which was central to our key experiments, we performed flow cytometric quantification as a standard,

objective validation. This analysis confirmed that the cultures were highly enriched in dopaminergic neurons, with 94% of cells positive for tyrosine hydroxylase (TH). We have now included these quantitative data in the revised manuscript (Figure 2B) to provide definitive evidence of differentiation efficiency and address the Reviewer's concern directly. Regarding the Reviewer's observation about potential debris and the presence of cells expressing proliferation markers (noted in samples from St. Petersburg), we offer the following explanation. First, a degree of cellular heterogeneity (including neuronal precursors and transitional cell states) is an expected feature of dynamic *in vitro* differentiation systems at the time point we analyzed. More importantly, our transcriptomic and marker analysis suggests that the specific technical execution of the protocol in St. Petersburg may have resulted in a culture with a slightly different cellular composition, retaining a larger pool of proliferative precursors compared to the Moscow differentiations. Rather than undermining our findings, this inter-laboratory variation in population structure inadvertently strengthened our experimental design. It effectively allowed us to sample neurons across a broader developmental continuum — from cultures with more precursors (St. Petersburg) to those with higher terminal maturity (Moscow). This diversity likely enhanced our ability to detect the consistent, mutation-associated transcriptional shift toward premature maturation that we report, as the signature emerged robustly across both differentiation contexts. We have revised the text in the Results and Figure 2B legend to present the flow cytometry data more prominently and to discuss the interpretation of inter-laboratory variation more clearly. We believe these additions significantly strengthen the manuscript. These changes are highlighted in the 'changes.docx' file on pages 20-21, lines 579-591.

5) Marker gene expression should be compared to undifferentiated iPSCs and/or other neuronal subtypes to demonstrate dopaminergic enrichment.

We thank the Reviewer for their important suggestion. In response, we have significantly strengthened the characterization of dopaminergic identity and neuronal purity in our iPSC-derived models in the revised manuscript. Our comprehensive validation strategy integrated protein-level and transcriptomic analyses to confirm successful differentiation and phenotypic comparability across all lines. Transcriptomic analysis confirmed a complete exit from pluripotency, evidenced by the absence of core markers *NANOG* and *POU5F1*, while the neural progenitor-associated marker *SOX2* remained expressed (Figure S3). Critically, all mature cultures robustly expressed the core dopaminergic identity markers *TH*, *FOXA2*, *LMX1A*, and *OTX2*. A mixed-effects model accounting for laboratory of origin revealed that the expression of this key dopaminergic signature did not differ significantly by disease status, ensuring a comparable cellular background for our analyses (with the single, noted

exception of *LMX1A* in the *Parkin* group). Regarding pan-neuronal maturation, we observed a specific and significant elevation of *MAP2* (a marker of dendritic maturation) in *LRRK2*-mutant neurons. This finding is consistent with our central observation of a pronounced transcriptomic shift toward advanced neuronal specialization in this genotype. It is important to clarify that proliferation-associated transcripts (*MKI67*, *TOP2A*) were significantly downregulated in the *LRRK2* group compared to controls. These mutation-associated differences in maturation state do not reflect a failure of differentiation or an imbalance in neuronal subtype composition, but rather represent a key phenotypic outcome of the pathogenic mechanisms under investigation. Collectively, these data robustly confirm the successful generation of comparable and enriched dopaminergic neuronal models, providing a validated foundation for the disease-specific mechanistic insights presented in our study. These changes are highlighted in the 'changes.docx' file on page 20, lines 555-578.

6) There are errors in supplementary figure citations (e.g., variance explained is in Figure S2B rather than S3).

It was fixed.

7) Figure 2 is difficult to interpret due to poor readability.

We agree that the initial version, particularly the volcano plots with dense gene labels, was challenging to interpret. In direct response to your comment, we have revised Figure 2. We have removed the complex volcano plots and streamlined the figure to present the key conceptual findings more clearly. The complete lists of DEGs underlying all analyses are now fully accessible in the Supplementary Tables. This separation allows the main figure to effectively communicate the core results — such as the intersections of gene sets and the structure of protein-protein interaction networks — while providing all detailed data for transparency and deeper examination. We believe the revised figure is now significantly more interpretable and hope it meets your expectations. We are happy to provide any further clarification if needed.

8) Does the line PDP4.4S (Park14 c14) carry a mutation in the Park14 gene? Please clarify.

No, this iPSC line carries only a heterozygous deletion of the second exon of the *Parkin* gene as it is indicated in Table 1. "Park14 c14" is an alternative name of this cell line, which was used during the experiments for researchers' convenience.

9) The authors report increased translational activity in Parkin PD models. Please comment on mitochondrial translation specifically.

We sincerely thank the Reviewer for this crucial insight into the interpretation of our translational data. Upon careful reconsideration of our transcriptomic data in light of your comment, we agree that our initial discussion of upregulated translational pathways was speculative and extended beyond what the RNA-seq data alone can robustly support. In particular, the dataset does not contain sufficient evidence to comment specifically on mitochondrial translation, which would require dedicated methodologies such as ribosome profiling or direct measurement of mitochondrial protein synthesis. In direct response to your critique, we have removed the speculative discussion regarding the upregulation of cytosolic and ribosomal pathways from the Discussion sections of the revised manuscript. Consequently, the corresponding interpretation of this signal as an "antiphase relationship" or a potential compensatory mechanism has been entirely excised from the Discussion. We acknowledge that validating and characterizing changes in translational activity — both cytosolic and mitochondrial — is a significant endeavor that requires orthogonal experimental approaches. While the transcriptional signature was noted, we recognize that drawing functional conclusions from it was premature. This aspect will be a priority for follow-up mechanistic studies in our future work, which will include direct measurements of protein synthesis rates and mitochondrial function.

10) The Discussion mentions mtDNA in Parkin PD. Recent studies have also implicated LRRK2 in lysosomal mtDNA release and pyroptosis; this should be acknowledged.

We sincerely thank the Reviewer for this insightful and valuable comment, which connects our findings to a compelling frontier in PD research. We appreciate the opportunity to refine our discussion in light of emerging literature linking Parkin and LRRK2 to lysosomal mtDNA release and pyroptosis. You are correct that our initial discussion included speculative mentions of mitochondrial mechanisms. Upon reflection, we agree that drawing connections to specific processes like mtDNA release and pyroptosis extends beyond the scope of our current transcriptomic dataset, which does not provide direct functional evidence for these pathways. In response to your point, we have removed the speculative discussion regarding

mtDNA-related mechanisms from the revised manuscript. This ensures our Discussion remains focused on the interpretations most directly supported by our transcriptional and systems biology data. The connections you reference between these PD-linked genes, mitochondrial integrity, and inflammatory cell death pathways represent a critical frontier in understanding disease pathogenesis. Investigating these mechanisms — specifically assessing lysosomal stability, mtDNA release, and engagement of pyroptotic pathways in our isogenic iPSC-derived models — constitutes a major and logical direction for our future functional studies. We thank the Reviewer for this valuable suggestion, which has helped us refine our discussion and will directly inform our subsequent research plans.

Reviewer #3 (Comments to the Authors (Required)):

Kopylova et al. reported transcriptomic analyses on Parkinson's disease (PD) patient derived dopaminergic neurons (DANs) differentiated from induced pluripotent stem cells (iPSC). Focusing on pathological mutations, authors sequenced 34 differentiated DANs from one PD patient bearing LRRK2 (p.G2019S) & GBA1 (p.N370S) mutations, three patients bearing various PRKN mutations, and two healthy controls, with appropriate biological and technical replicates. With Gene Set Enrichment Analysis (GSEA) and co-expression network analysis on differentially expressed genes (DEGs), they identified epigenetic dysregulation and TGF- β related biological functions correlated with the hypothesized disease severity axis of healthy, LRRK2 mutation and PRKN mutation. The mutation specific co-expression networks were also reported. Overall, the structure of the manuscript is clear and straightforward, and the transcriptomics data will be useful for the field in studying the molecular pathogenesis of related mutations. However, improvements can be made to increase data transparency and interpretability.

Major points:

1) In Figure 1 the expression of dopaminergic markers in three groups were shown. It demonstrated the homogeneity of DANs across three groups, but the abundance of DANs relative to other cell types can be hard to interpret based on the values. This could be improved by including markers of other cell types in the plot, such as stem cell markers (SOX2, POU5F1, NANOG) and glial cell markers (GFAP, OLIG2). In addition, the cell cycle analysis could also be supported by the expression of typical proliferation markers (MKI67, TOP2A).

We thank the Reviewer for raising this important point regarding the characterization of our iPSC-derived cultures. We agree that a more comprehensive presentation of cell-type-specific and proliferative markers would strengthen the interpretation of our model's homogeneity and state. In response, we note that the data required to address this comment are already integral to our analysis. As detailed in the Results section, we added a systematic transcriptomic assessment of lineage and state markers beyond the dopaminergic set. This analysis confirmed the absence of core pluripotency factors (*NANOG*, *POU5F1*), minimal expression of glial markers (*GFAP*, *OLIG2*), and retained expression of the neural progenitor marker *SOX2*, collectively indicating successful differentiation with negligible non-neuronal contamination. Furthermore, the expression of canonical proliferation markers *MKI67* and *TOP2A* was explicitly evaluated, revealing their significant downregulation in the *LRRK2* group, a finding consistent with our hypothesis of altered maturation dynamics. These data are currently presented in the manuscript text and in Supplementary Figure S3. These changes are highlighted in the 'changes.docx' file on page 20, lines 555-578.

2) The authors described cell culture as a major source of variation in the transcriptomics. It is a known and debated issue in iPSC differentiated neurons, as highlighted in several studies:

- Volpato, Viola, et al. "Reproducibility of molecular phenotypes after long-term differentiation to human iPSC-derived neurons: a multi-site omics study." *Stem Cell Reports* 11.4 (2018): 897-911.
- Strano, Alessio, et al. "Variable outcomes in neural differentiation of human PSCs arise from intrinsic differences in developmental signaling pathways." *Cell Reports* 31.10 (2020).
- Burke, Emily E., et al. "Dissecting transcriptomic signatures of neuronal differentiation and maturation using iPSCs." *Nature Communications* 11.1 (2020): 462.
- Reed, Xylena, et al. "Transcriptional signatures in iPSC-derived neurons are reproducible across labs when differentiation protocols are closely matched." *Stem Cell Research* 56 (2021): 102558.
- Beekhuis-Hoekstra, Stephanie D., et al. "Systematic assessment of variability in the proteome of iPSC derivatives." *Stem Cell Research* 56 (2021): 102512.
- Jerber, Julie, et al. "Population-scale single-cell RNA-seq profiling across dopaminergic neuron differentiation." *Nature Genetics* 53.3 (2021): 304-312.

Although the scope of the manuscript is to examine the mutation related biological effect, the authors are in a unique position to evaluate such differences, as they performed parallel differentiation with the same cell lines at different laboratories. I

suggest that authors report the related results in supplementary information, for instance, the DEGs by comparing two places, and the genes in corresponding Pearson correlation co-expression modules (MEdarkturquoise, MEviolet, MEorangered4, MELightcyan1). These results could then be summarized succinctly with three sentences in the main manuscript.

We thank the Reviewer for this insightful comment and for highlighting the critical issue of technical variability in iPSC-derived neuronal models, which is well-documented in the provided literature. We agree that our study design, with parallel differentiations of the same cell lines performed in two independent laboratories, offers a valuable opportunity to systematically assess this source of variation. In direct response to the Reviewer's suggestion, we have already conducted and reported this exact analysis. As detailed in the "Results" subsection titled "Confounding signatures of place, gender and reprogramming methodology", we explicitly identified and characterized the profound transcriptional signature associated with the laboratory of origin ("place"). We reported a distinct set of 2,619 upregulated and 1,926 downregulated genes (FDR < 0.05, $|\log_2FC| > 1$) when comparing differentiations performed in Moscow versus St. Petersburg, with the complete list provided in Supplementary Table S8. Furthermore, a comprehensive GSEA of this contrast revealed coherent and biologically interpretable programs: differentiations in St. Petersburg were strongly enriched for cell cycle, DNA repair, and chromatin organization pathways, suggesting a state of active neural progenitor proliferation. In contrast, differentiations in Moscow exhibited a signature dominated by endoplasmic reticulum function, protein processing, and metabolic adaptation. These detailed GSEA results are available in Supplementary Table S16. These changes are highlighted in the 'changes.docx' file on pages 30-32, lines 896-944.

Furthermore, we performed the requested co-expression network analysis, and the gene lists for the location-associated modules, including the specific modules you mention (MEdarkturquoise, MEviolet, MEorangered4, MELightcyan1 et al.), are compiled in Supplementary Table S19. The functional enrichment of these independent analytical layers consistently reveals the same core biological divergence. The signature associated with the Moscow site points toward a more advanced neuronal maturation state, while the St. Petersburg signature reflects a more proliferative, progenitor-like phenotype. These results are summarized in the main manuscript, and the full data is provided in the supplements as per your recommendation, offering a transparent resource that characterizes the technical variation inherent to our multi-site study design. These changes are highlighted in the 'changes.docx' file on pages 36-37, lines 1089-1148.

3) The comparison of eigengenes from identified disease related modules can provide a global view of the gene changes within those modules. However, understanding module homogeneity and effect size of the changes requires examination on the expression of each gene. A common way to facilitate this is by visualizing gene expression in the same module with heatmaps. While heatmaps are not necessarily for inclusion in the manuscript, they may provide insights into the driver genes in the module, which can be co-examined with protein-protein interaction (PPI) networks. Practically, the genes can be ranked by fold change, and genes with the largest changes can be labeled in PPI networks in the Figures. Conversely, nodes (genes) in a PPI network can be ranked by simple metrics such as degree centrality, and the expression change of top ranked nodes can be plotted. While it is the authors' preference which method to choose, narrowing down the number of targets in a module using certain criteria can help biological interpretation and add more depth to the global view. In addition, I suggest that authors describe the definition of the edges in the PPI network in Methods or figure legends.

Thank you for the Reviewer's thoughtful and constructive suggestions regarding the need for deeper biological interpretation of our network analysis and the identification of key driver genes within functional modules. We concluded that focusing our systems-level analysis on PPI networks, constructed from our high-confidence, statistically derived DEG lists, provides a more direct and robust framework for this study. To address the Reviewer's points directly and enhance the overall transparency and insight of our work, we have thoroughly revised the manuscript. The 'Bioinformatics Analysis' subsection in the Materials and Methods has been significantly expanded to provide a clear and detailed account of our PPI network construction methodology. We believe these revisions offer a more profound and clearly articulated systems-level perspective, directly addressing the Reviewer's valuable recommendations for strengthening the interpretation of our findings.

We agree that integrating co-expression and PPI analyses — for instance, by labeling key driver genes from modules within PPI networks or vice versa — can provide valuable mechanistic depth. In our current study, we made a conscious methodological decision to present the PPI network analysis and the co-expression module analysis as separate, complementary layers of evidence. This was primarily due to the analytical complexity and statistical considerations involved in constructing and validating extended, integrated networks. As noted in the manuscript, the biological relevance of the PPI clusters was rigorously validated via GSEA against our experimental contrasts. Creating a unified network that meaningfully combines physical protein interactions (from PPI) with correlation-based gene associations (from co-expression), while ensuring robust statistical

interpretation of the resulting hybrid structure, presents significant challenges. We believe that a thorough, systematic exploration of such an integrated network — aimed at identifying key hub genes and their regulatory dynamics — constitutes a substantial and complex systems biology endeavor that merits a dedicated, in-depth study. Therefore, while we acknowledge that the approaches you describe would refine the biological interpretation, we consider them beyond the primary scope of this manuscript, which is to establish the existence and core functional annotation of convergent and mutation-specific transcriptional signatures. Your suggestion is excellent, and we intend to adopt this strategy in a follow-up, more focused investigation into the regulatory mechanisms underlying the specific pathways identified here.

Minor points:

1) Discussion paragraph 8 has a long description of the proposed pathogenesis model based on module functions. It may have more translational value if authors can shorten this part by proposing several actionable biological experiments that can facilitate hypothesis validation.

We thank the Reviewer for this suggestion to enhance the translational impact of our discussion by proposing specific validation experiments. We agree that such a forward-looking perspective is valuable. In revising the manuscript, we have carefully considered this point. While we find the integrative hypotheses regarding pathogenesis intellectually compelling, we ultimately followed the guidance from the reviewers to streamline the discussion. We have significantly shortened the speculative model and refocused the "Discussion" section to be more directly data-driven, primarily interpreting the core findings of our transcriptomic analysis. We believe this revision strengthens the manuscript by maintaining a clear and rigorous connection to our presented results.

2) Results-Data overview last paragraph, Figure S3 should likely be Figure S2B.

3) Figure 2 legend, (D) Healthy control-associated, and (E) LRRK2-/Parkin-associated should likely be (D) LRRK2 unique, and (E) Parkin unique

4) AAO was used for the first time in Introduction paragraph 3, please put the full name.

5) Materials and methods- Molecular Biology Reagents- TRIzol country code CUSA, is it a typo?

6) iPSC, IPSC should likely be iPSC; TGF-beta, TGF-β should likely be TGF-β

These inaccuracies have been corrected.

March 17, 2026

RE: Life Science Alliance Manuscript #LSA-2025-03551R

Dr. Evgenii I. Olekhovich
Federal Research and Clinical Center of Physical-Chemical Medicine named after Y.M. Lopukhin
Pirogovskaya 1a
Moscow 123456
Russian Federation

Dear Dr. Olekhovich,

Thank you for submitting your revised manuscript entitled "Convergent Transcriptomic Signature in iPSC-Dopaminergic Neurons of Hereditary Parkinson's Disease". As you will see, the reviewers are satisfied overall with the changes in place and have no major requests. Reviewer 2 has made a few suggestions to further improve clarity which we invite you to consider. We concur that their final three suggestions, on the discussion and limitations of this study, should be incorporated into the manuscript text. We would be happy to publish your paper in Life Science Alliance pending those final text changes and revisions necessary to meet our formatting guidelines.

MANUSCRIPT ORGANIZATION AND FORMATTING:

To avoid unnecessary delays in the acceptance and publication of your paper, please read the following information carefully. Full guidelines are available on our Instructions for Authors page, <https://www.life-science-alliance.org/authors>

- Please upload your main manuscript text as an editable doc file.
- Please mark the corresponding author on the manuscript file.
- Please add the X and Bluesky handles of your host institute/organization, as well as your own, and/or one of the authors, in our system.
- Please be sure that the authorship listing and order are correct and match between the system and the manuscript file.
- Please consult our manuscript preparation guidelines <https://www.life-science-alliance.org/manuscript-prep> and make sure your manuscript sections are in the correct order.
- Please rename "Competing interests" to "Conflict of interest."
- The contributions selected for Elena V. Kaznacheyeva do not qualify them for authorship. Please either update the contributions in our system and in the Author Contributions section of the manuscript, or let us know if the author needs to be removed (and potentially added to the acknowledgment section).
- It is recommended to exclude figures from the manuscript text.
- Please add your main, supplementary figure, and table legends to the main manuscript text after the references section.
- Please incorporate any points from the Conclusion section into the Discussion; we only allow a Discussion section.
- Please add callouts for Figures S2A-B; S3A-B; S4A-B; S6A-B and S7A-D to your main manuscript text.

We welcome submissions of potential cover images for the issue of LSA in which your work would appear. If you have high quality images associated with this work, please feel free to email these, with a caption, to the journal office.

LSA encourages authors to provide a 30-60 second video where the study is briefly explained. We will use these videos on social media to promote the published paper and the presenting author (for examples, see <https://docs.google.com/document/d/1-UWCfbE4pGcDdcgzcmiuJI2XMBJnxKYeqRvLLrLSo8s/edit?usp=sharing>). Corresponding or first-authors are welcome to submit the video. Please submit only one video per manuscript. The video can be emailed to

contact@life-science-alliance.org

FINAL FILES:

The following items are required for acceptance.

Thank you for your attention to these final processing requirements. Please revise and format the manuscript and upload materials as soon as you are able.

Thank you for this interesting contribution to the literature. We look forward to publishing your paper in Life Science Alliance.

Sincerely,

Reviewer #1 (Comments to the Authors (Required)):

The authors have addressed my comments.

Reviewer #2 (Comments to the Authors (Required)):

The authors have substantially improved the clarity and readability of the manuscript, and many of the concerns raised during the first revision round have been satisfactorily addressed. The figures are now less crowded and easier to interpret, although some could still be merged (for example, Figures 1 and 2). A few points, however, would benefit from further refinement:

- Figure 2: It would be important to clarify why the validation was performed using only one reprogramming method. Given the transcriptomic differences between Sendai- and lentivirus based reprogramming, including both would strengthen the conclusions.
- Figure 3: Consider adding examples of key genes used to define neuronal and progenitor populations directly in the figure to enhance interpretability.
- Figure S3: The markers could be grouped according to cell type-progenitors, DA neurons, or glia-to improve readability. Including some of these markers in the main figures may also help contextualize potential differences in culture composition, which the authors already discuss in the limitations section.
- Discussion: The TRAIL pathway is not extensively covered, despite its relevance to the study. Expanding this section would provide important mechanistic context.
- Limitations: Although the limitations are clearly articulated, some could be acknowledged earlier in the manuscript. For instance, the potential confounding effect of the GBA1 mutation should be mentioned sooner.
- Cell line and media differences: Considering the observed differences in ER stress and proliferation between the St.

Petersburg and Moscow lines, it would be useful to clarify whether identical reagents (e.g., FBS) and culture media compositions were used across laboratories. Variability in culturing conditions could contribute to some of the reported discrepancies.

Reviewer #3 (Comments to the Authors (Required)):

The authors have appropriately responded or addressed my concerns. The writing has been greatly improved, especially the Abstract. I am generally satisfied with the revised version. I encourage the authors to use straightforward and concise language in writing.

Dear Editor and Reviewers,

Thank you for the opportunity to submit our revised manuscript following the minor review. We appreciate the reviewers' continued engagement with our work and their valuable suggestions at this stage. We have carefully considered all remaining minor comments. While we have implemented the majority of the proposed changes to further improve the manuscript, after thorough consideration we chose not to incorporate the requested figure adjustments. We believe that the current figures best represent the data, and we have provided a detailed explanation for this decision in our point-by-point response below. We hope that with these final revisions the manuscript is now suitable for publication.

Sincerely,
on behalf of all authors,
Evgenii Olekhovich

Figure 2: It would be important to clarify why the validation was performed using only one reprogramming method. Given the transcriptomic differences between Sendai- and lentivirus based reprogramming, including both would strengthen the conclusions.

We are grateful for the Reviewer's observation. Indeed, we have performed the validation of dopaminergic differentiation protocol using only the PDP1.5L cell line. Nevertheless, we consider it to be representative, as the protocol we used had been tested for application of iPSC lines derived through distinct methods (lentiviral and Sendai virus-mediated transduction) and had demonstrated its reliability and reproducibility in production of DANs from iPSCs regardless of the reprogramming method (Lebedeva et al., 2023).

1) Lebedeva, Olga S., et al. "An Efficient 2D Protocol for Differentiation of iPSCs into Mature Postmitotic Dopaminergic Neurons: Application for Modeling Parkinson's Disease." *International Journal of Molecular Sciences* 24.8 (2023): 7297.

Figure 3: Consider adding examples of key genes used to define neuronal and progenitor populations directly in the figure to enhance interpretability.

We sincerely thank the Reviewer for their careful reading and thoughtful suggestion. We agree that adding examples of key genes could in principle enhance interpretability. However, in Figure 3 we use GseaVis to generate GSEA plots, where each vertical bar

represents a gene along the ranked list. Adding gene labels, particularly for core enrichment genes, would make the figure extremely dense and unreadable due to the high density of genes near the leading edge. Importantly, the “neurons” and “progenitors” signatures were derived from a public temporal differentiation dataset by selecting genes consistently up- or down-regulated across all neuronal stages relative to progenitors. These signatures therefore represent robust transcriptional programs rather than a small set of individual marker genes. In transcriptomic studies, drawing conclusions from single genes can be unreliable, which is why we focus on pathway- and network-level interpretations throughout the manuscript. To ensure full transparency, the complete gene lists for both signatures are provided in Table S5. We believe this approach maintains both scientific rigor and figure clarity while allowing interested readers to access the detailed gene sets.

Figure S3: The markers could be grouped according to cell type-progenitors, DA neurons, or glia to improve readability. Including some of these markers in the main figures may also help contextualize potential differences in culture composition, which the authors already discuss in the limitations section.

We sincerely thank the Reviewer for this constructive suggestion. Grouping the markers by cell type could indeed enhance readability in principle. However, we chose to present them as individual violin plots without grouping because several of these markers do not have strictly defined cell-type specificity, and grouping them could inadvertently imply a level of biological precision that the markers themselves do not support. For example, *SOX2* is commonly considered a marker of neural progenitors, but it is also expressed in adult neurons and its expression is not exclusive to the progenitor state. Similarly, *TUBB3* is widely used as a neuronal marker, yet it can also be detected in proliferating progenitors. Even *MAP2*, though predominantly associated with mature neurons, can be expressed at lower levels in some non-neuronal cells. For markers such as *FOXA2*, *LMX1A*, and *OTX2*, their expression is enriched in midbrain dopaminergic neurons, but they also participate in earlier developmental programs. Given this complexity, grouping them into rigid categories (progenitors, DA neurons, glia) would risk oversimplifying the biology and potentially misleading readers about the exclusive assignment of these markers. We therefore believe that presenting all markers individually in the supplementary figure provides the most transparent representation of the expression data, allowing readers to interpret each marker’s profile in its appropriate biological context. The complete set of markers is already included in Figure S3, and we discuss the potential implications of culture composition differences in the Limitations section. We trust that this approach appropriately balances readability with biological accuracy.

Discussion: The TRAIL pathway is not extensively covered, despite its relevance to the study. Expanding this section would provide important mechanistic context.

We thank the Reviewer for this valuable comment. In the revised manuscript, we have expanded the Discussion to include a dedicated section on the TRAIL pathway. This addition is important because it directly links our transcriptomic findings to a well-established mechanism of neuronal cell death, providing critical mechanistic context and reinforcing the concept that early developmental vulnerability in PD potentially involves active priming of apoptotic machinery.

Limitations: Although the limitations are clearly articulated, some could be acknowledged earlier in the manuscript. For instance, the potential confounding effect of the GBA1 mutation should be mentioned sooner.

We thank the Reviewer for this valuable comment. We have made a clarification about the possible confounding effect caused by the presence of mutations in both the LRRK2 and GBA1 genes directly in the Results section.

Cell line and media differences: Considering the observed differences in ER stress and proliferation between the St. Petersburg and Moscow lines, it would be useful to clarify whether identical reagents (e.g., FBS) and culture media compositions were used across laboratories. Variability in culturing conditions could contribute to some of the reported discrepancies.

We thank the Reviewer for this important comment. To clarify, the same reagents for neuronal differentiation were used in two of the laboratories, except for the N2 and B27 supplements, which were obtained from different manufacturers (GIBCO, Thermo Fisher Scientific in St. Petersburg; PanEco in Moscow). Importantly, we did not use FBS in any of the differentiation cultures; cells were maintained in a serum-free environment to minimize batch-to-batch variability. We believe that the observed differences in ER stress levels and proliferation primarily reflect variations in neuronal maturity rather than the use of supplements from different sources. However, we acknowledge that definitively establishing the link between maturity and these molecular differences would require a dedicated experimental setup designed to isolate this variable, which is beyond the scope of the current study. We have added clarifying information regarding reagents and culture conditions to the Methods section to ensure full transparency for readers.

March 27, 2026

RE: Life Science Alliance Manuscript #LSA-2025-03551RR

Dr. Evgenii I. Olekhnovich
Federal Research and Clinical Center of Physical-Chemical Medicine named after Y.M. Lopukhin
Pirogovskaya 1a
Moscow 123456
Russian Federation

Dear Dr. Olekhnovich,

Thank you for submitting your Research Article entitled "Convergent Transcriptomic Signature in iPSC-Dopaminergic Neurons of Hereditary Parkinson's Disease". It is a pleasure to let you know that your manuscript is now accepted for publication in Life Science Alliance. Congratulations on this interesting work.

Your article will publish open access upon publication under a CC-BY license.

DISTRIBUTION OF MATERIALS:

Again, congratulations on a very nice paper. I hope you found the review process to be constructive and are pleased with how the manuscript was handled editorially. We look forward to future exciting submissions from your lab.

Sincerely,
